# Robust prediction of synthetic gRNA activity and cryptic DNA repair by disentangling cellular CRISPR cleavage outcomes

Stephan Riesenberg [1] ✉, Philipp Kanis [1], Rosa Karlic [2] & Tomislav Maricic [1]

The ability to robustly predict guide RNA (gRNA) activity is a long-standing goal for CRISPR applications, as it would reduce the need to pre-screen gRNAs. Quantification of formation of short insertions and deletions (indels) after DNA cleavage by transcribed gRNAs has been typically used to measure and predict gRNA activity. We evaluate the effect of chemically synthesized Cas9 gRNAs on different cellular DNA cleavage outcomes and find that the activity of different gRNAs is largely similar and often underestimated when only indels are scored. We provide a simple linear model that reliably predicts synthetic gRNA activity across cell lines, robustly identifies inefficient gRNAs across different published datasets, and is easily accessible via online genome browser tracks. In addition, we develop a homology-directed repair efficiency prediction tool and show that unintended large-scale repair events are common for Cas9 but not for Cas12a, which may be relevant for safety in gene therapy applications.

Targeted cleavage of genomic DNA by a complex of gRNA and CRISPR–Cas9 results in a DNA double-stranded break (DSB). This DSB can trigger p53-dependent cell death[1,2] or cellular DNA repair that can result in modification of the target locus and thus genome editing (Fig. 1a). Simultaneous cleavage of more than one target, e.g., by off-target cleavage[3], potentiates the p53-dependent DNA damage response[4]. The most efficient repair pathways are non-homologous end joining (NHEJ) and microhomology-mediated end joining (MMEJ) which can both result in short insertions or deletions of a few bases (indels). To date, PCR amplification of the target locus for quantification of indels has been the conventional metric for quantifying gRNA efficiency[5–8]. However, there are DNA repair events, which we will refer to as 'cryptic repair', that cannot be quantified by short-range PCR amplification. For example, end joining pathways can also result in large-scale DNA modifications that cannot be detected by standard short-range amplification, such as deletions of several hundred bases, translocations, inversions, and chromosome arm loss[9–14]. Also, NHEJ but not MMEJ can result in perfect repair of the DSB without any modifications[15]. In addition to NHEJ and MMEJ, cells can repair DSBs by homologous recombination (HR), which accurately copies the sister chromatid, or by related homology-directed repair (HDR), when an exogenous DNA donor carrying mutations of interest is provided[16]. Consequently, conventional quantification of indels alone is blind to many other possible outcomes (e.g., cell death, perfect repair, large-scale repair) that are informative of successful gRNA cleavage of the target DNA. To use only indels as a measure of gRNA activity (its ability to guide the Cas protein to a specific genomic location to make a DSB) may therefore bias the scoring and prediction of true gRNA activity (Supplementary Fig. 1).

For any CRISPR application, it is critical to select a gRNA that results in efficient target cleavage or binding which is a prerequisite for efficient downstream genome or transcriptome manipulation. gRNAs can be transcribed from a DNA template or produced by chemical synthesis. The latter 'synthetic' gRNAs are usually used for ribonucleoprotein (RNP) delivery. Unfortunately, existing Cas9 gRNA efficiency prediction models only imperfectly predict gRNA efficiencies[17]. In addition to suboptimal approximation of gRNA activity by indels[6–8] or phenotype depletion[18], previous models were trained on datasets where gRNA production was subject to sequence biases during transcription from the U6 (in vivo) or T7 promoter (in vitro). For example,

[1]Department of Evolutionary Genetics, Max Planck Institute for Evolutionary Anthropology, Leipzig, Germany. [2]Bioinformatics Group, Division of Molecular Biology, Department of Biology, University of Zagreb, Zagreb, Croatia. ✉e-mail: stephan_riesenberg@eva.mpg.de

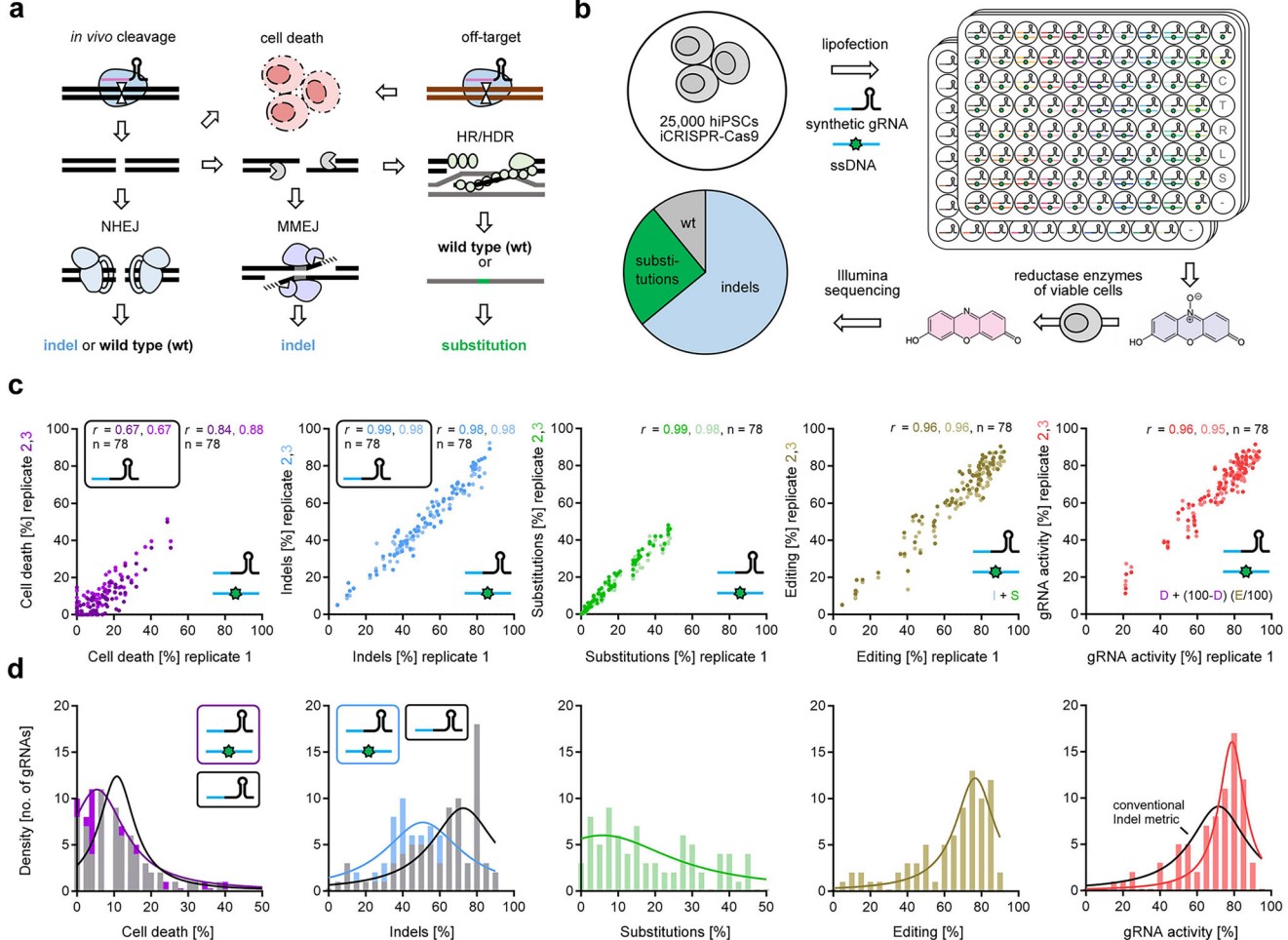

**Fig. 1 | Cellular synthetic gRNA CRISPR-Cas9 cleavage outcome screen.**
**a** Schematic of potential cellular cleavage outcomes after a CRISPR induced DSB. Cells can die as a consequence of the DSB or repair it by competing pathways (NHEJ, MMEJ, HR/HDR), which can result in perfect repair to the wild type sequence, insertions or deletions (indels), or targeted substitutions when a DNA donor is provided. **b** Chemically synthesized (synthetic) gRNA screen design. 409B2 hiPSCs expressing Cas9 were lipofected with synthetic oligonucleotides in one 96 well per target (n = 78 targets). Cell survival was measured by a resazurin fluorescence assay, followed by subsequent DNA isolation and Illumina sequencing of target PCR amplicons. **c** Scatter plots of replicates for percentage of cell death in purple, indels in blue, substitutions in green, editing (sum of indels and substitutions) in gold, and

in vivo gRNA activity (quantification of cell death and editing) in red, when editing with gRNA and DNA donor. Darker and brighter dots indicate the correlation of replicate one with respect to replicate two and three, respectively. Pearson's r for correlation (two-tailed) of independent biological replicates (n = 3) are stated (black frame for editing without DNA donor). All p values are <0.0001.
**d** Histograms showing the density of the distribution of gRNAs that result in different mean (n = 3) percentages of cell death, indels, substitutions, editing, and gRNA activity, when editing with gRNA and DNA donor. For cell death and indels, gray bars represent the distribution for editing without DNA donor. Lines show the respective Lorentzian distributions (black for editing without DNA donor, colored for editing with DNA donor). Source data are provided as a Source Data file.

polymerases can favor transcription of gRNAs based on sequence composition and can introduce 5'- and 3'- inhomogeneity[19–22], with the result that unequal amounts and/or errors of transcribed gRNAs in screens can bias the inference of cleavage efficiency. Furthermore, Pol III can terminate transcription of gRNAs with T-stretches of only two thymines[23]. Inevitably, sequence features responsible for gRNA transcription bias have been incorporated into prior prediction models, thus biasing against those responsible for gRNA catalytic activity. Consequently, transcription bias-free chemically synthesized 'synthetic' gRNA could help to reveal gRNA target sequence features that are truly responsible for gRNA activity.

Here, we disentangle cellular CRISPR–Cas9 cleavage outcomes in human cells after cleavage by synthetic gRNA and reveal sometimes strong discrepancy between gRNA activity and genome editing efficiency. Compared to transcribed gRNAs, we find that synthetic gRNAs have fewer sequence features that are detrimental to efficient genome editing. This finding allowed us to reliably predict synthetic Cas9 gRNA activity as well as HDR efficiency in different cell types. In addition, we

show that Cas9 and Cas12a have different propensities for cellular cleavage outcomes.

## Results

### Efficiency metrics screen of synthetic gRNAs
To measure different cellular outcomes after a Cas9-induced DSB, we lipofected synthetic gRNAs (hybridized crRNA:tracrRNA) with or without corresponding single-stranded DNA donors into 409-B2 hiPSCs expressing Cas9 after doxycycline treatment (iCRISPR)[24] (Fig. 1b). Two days after lipofection, we measured cell death using a resazurin assay[25] and then extracted DNA for targeted PCR followed by Illumina sequencing to quantify indels and targeted substitutions. Resazurin fluorescence for lipofection with a gRNA designed not to have a target in the human genome was considered as 100% cell survival. We obtained highly reproducible efficiency metrics for 78 gRNAs when editing hiPSCs (Fig. 1c). These are (I) percentage of 'indels' generated when no DNA donor is used (conventional metric), (II) percentage of 'genome editing', which is the sum of indels and targeted

substitutions when a DNA donor is provided, and (III) CRISPR cleavage-induced 'cell death'. The latter two are combined into (IV) 'in vivo gRNA activity' (Supplementary Data 1).

To assess gRNA activity independent of cellular DSB responses, we used gRNAs for in vitro cleavage of the respective double-stranded DNA target containing 60/30 bp human genome sequence 5'/3' of the cut (Supplementary Fig. 2a). After incubation with Cas9 RNP we quantified the cutting efficiency by size separating capillary electrophoresis and obtained (V) percentage of 'in vitro gRNA cleavage' for a subset of 56 gRNAs (Supplementary Fig. 2b, c) (Supplementary Data 1).

### Conventional indel frequency metric can strongly underestimate gRNA activity

To evaluate the extent of different cellular cleavage outcomes, we plotted the efficiency distributions of cell death, indels, genome editing, and in vivo gRNA activity (Fig. 1d). CRISPR-induced cell death can reach up to 50%. The mean indel frequency when no DNA donor is used is 59% with a range of 6–90%, and one-fifth of gRNAs generate less than 40% indels. Combining cell death and editing results in that a striking

93% of gRNAs have at least around 40% activity (mean 68%). The mean in vitro gRNA cleavage is 32% with a range of 11–68% (Supplementary Fig. 2c). In vitro cleavage defines the lower limit of in vivo gRNA activity (Supplementary Fig. 2d), and correlates with in vivo activity, but not with indels or other efficiency metrics (Supplementary Fig. 2e), highlighting the shortcomings of indel quantification alone as a gRNA activity proxy. Also, indels and editing are negatively correlated with cell death, suggesting that when many cells die due to DSBs, only a few can repair the DSBs via indel formation. Non-targeting gRNAs do not reduce cell survival (Supplementary Fig. 2f), while cell death introduced by targeting gRNAs can result in big differences of indels and in vivo gRNA activity (Supplementary Fig. 2g).

### Sequence features influencing synthetic in vivo gRNA activity

We next tested the ability of published prediction tools trained on data from transcribed gRNAs[7,18,26,27] (Doench 2016 score, Rule Set 3 score, DeepHF score, CRISPRscan score), as well as an undisclosed proprietary tool trained on data from synthetic gRNAs (IDT on-target score) to predict both indels and in vivo gRNA activity of our screen (Fig. 2a). All

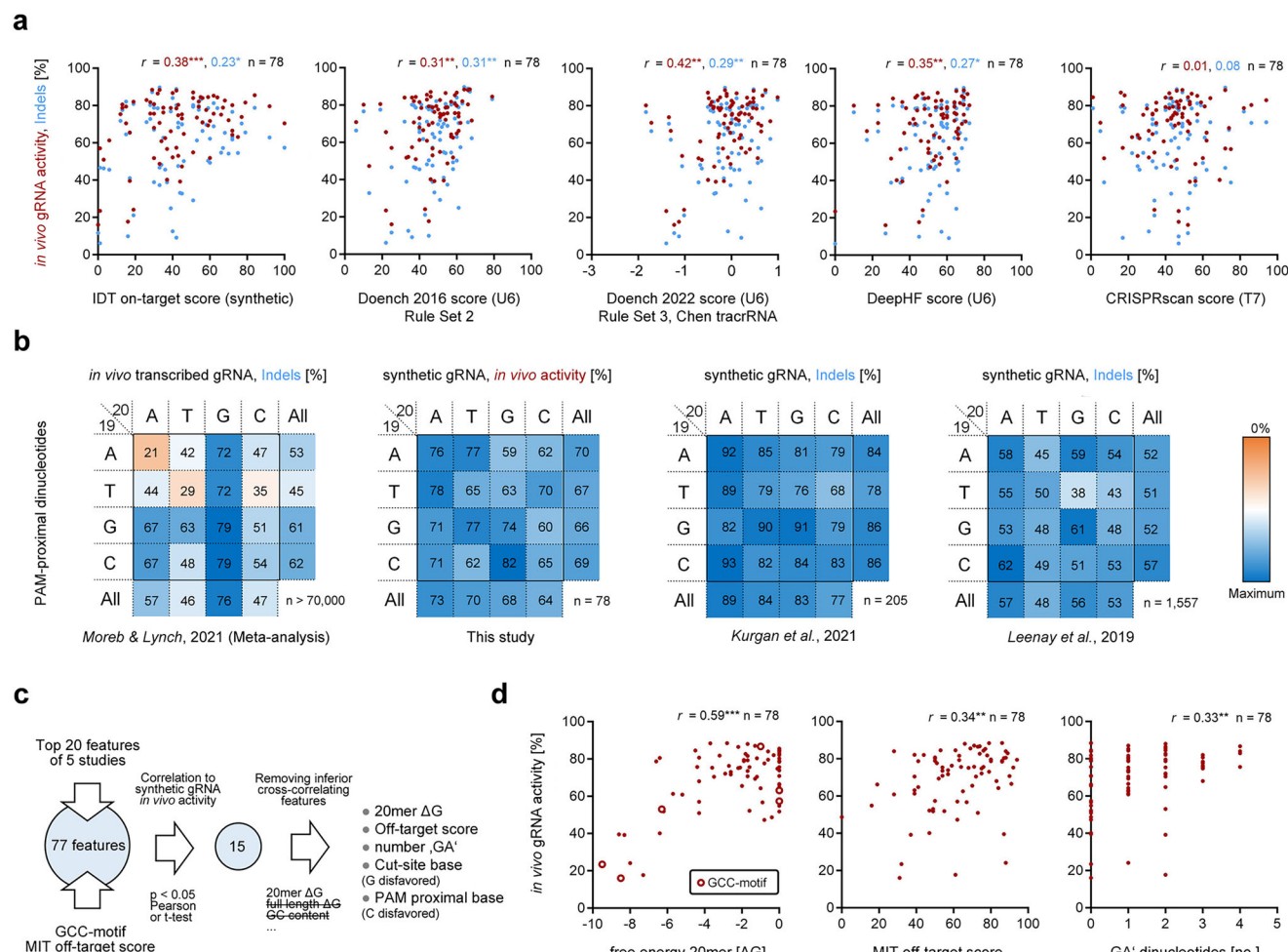

**Fig. 2 | Features influencing synthetic gRNA activity. a** Scatter plots of indels (blue) and in vivo gRNA activity (red) of our synthetic gRNA screen and scores from different gRNA efficiency prediction tools trained on synthetic gRNAs (undisclosed IDT score), in vivo U6 promoter transcribed gRNA[7,18,27], and in vitro T7 promoter transcribed gRNAs[26]. Pearson's r (two-tailed) is stated (*$p \leq 0.05$, **$p \leq 0.01$, ***$p \leq 0.001$). Each dot represents the mean of independent biological replicates (n = 3) for one gRNA. **b** Heatmaps of mean gRNA efficiency metrics for all possible PAM-proximal dinucleotide combinations (positions 19 and 20 of the gRNA). The left panel shows that efficiency from a meta-analysis over 70,000 in vivo transcribed gRNAs[53] is highly influenced by PAM-proximal nucleotide composition. In

contrast, synthetic gRNAs in our screen and other studies[39,40] show limited influence of PAM-proximal nucleotides. **c** Flowchart of gRNA feature selection and identification of predictive features for synthetic in vivo gRNA activity. **d** Scatter plots of in vivo gRNA activity and free energy of the 20nt gRNA sequence, MIT off-target score[33], and number of 'GA' dinucleotides. Pearson's r of the correlation (two-tailed) is stated (**$p \leq 0.01$, ***$p \leq 0.001$). Each dot represents the mean of independent biological replicates (n = 3) for one gRNA. gRNAs with a PAM-proximal GCC motif previously described to be detrimental for cleavage[23] are depicted as white filled circle in the free energy panel. Source data are provided as a Source Data file.

tools provide far from a satisfactory prediction, but the Rule set 3[27] correlates best with in vivo gRNA activity (Pearson's $r = 0.42$ with Chen tracrRNA, $r = 0.31$ with Hsu tracrRNA) and Rule set 2 (Doench 2016 score)[18] correlates best with indels ($r = 0.31$). Most of published on-target models were trained with transcribed gRNAs that use the tracrRNA as implemented in Hsu et al.[28] Chen et al. modified the Hsu tracrRNA with a flipped T to A substitution and compensatory A to T substitution, to disrupt the run of four thymidines that can trigger RNA polymerase III termination, and an extension of five base pairs in the tetra-loop to stabilize the gRNA structure[29,30]. CRISPRscan did not correlate with our data.

While G20 is the most robust feature found to be beneficial for efficiency of transcribed gRNAs[7,8,18,31], it has no impact on synthetic gRNA activity in our screen as well as in other datasets where synthetic gRNAs were used (Fig. 2b). Contrary to synthetic gRNAs, efficiency metrics of in vivo transcribed sgRNAs are highly dependent on the composition of PAM-proximal dinucleotides. Of note, the dataset for transcribed sgRNAs is more than 40-fold larger than the biggest synthetic gRNA dataset. Due to comparatively small sample sizes for each dinucleotide in studies using synthetic gRNAs, we cannot exclude potential moderate effects of PAM-proximal dinucleotides on synthetic gRNA efficiency.

To find features responsible for in vivo synthetic gRNA activity, we tested the predictive power of the top 20 features from each of six prediction tools trained on data generated from transcribed gRNA[6–8,18,26,31]. We also evaluated the impact of PAM proximal GCC, which has been described to be detrimental for cleavage[23,32], as well as likeliness of off-targets inferred from the MIT specificity score[33]. Of these combined 77 features, only 15 features correlated with synthetic in vivo gRNA activity (Fig. 2c). Removing cross-correlating features with inferior correlation to in vivo gRNA activity reduced significant contributing gRNA features from 15 to five: free energy of the spacer sequence, number of GA, MIT score, as well as nucleotide composition 5′ of the cut site (G17 disfavored) and directly before the protospacer adjacent motif (PAM) (C20 disfavored). Free energy, number of GA dinucleotides, and the MIT score each alone already show a comparable or better predictive performance for in vivo synthetic gRNA activity than the best performing tools mentioned above (Fig. 2d). Interestingly, we find that gRNAs with a PAM-proximal GCC trinucleotide are only inefficient when they have a very negative spacer free energy, while those with more positive free energy are efficient. This indicates that not the GCC motif itself, but its sequence dependent impact on free energy is the responsible mechanism for low activity of these gRNAs.

## Sequence features influencing HDR efficiency

Since our screen used DNA donors, which have same length homology arms and contain all possible nucleotide transitions adjacent to the related cut site in the genome, we can use the same screening data to also interrogate DNA donor sequence features influencing HDR efficiency. HDR efficiency, as well as the ratio of HDR-dependent intended edits to all genome editing events that differ from the wild-type sequence, which we refer to as 'HDR share', is highly correlated to free energy of the single stranded DNA donor ($r = 0.57$), but not to number of GA which we showed influences gRNA efficiency ($r = 0.01$) (Supplementary Fig. 3a). We further find that the type of nucleotide change can strongly influence HDR share. A change from cytosine to thymine is more than 2.5-fold more efficient than a change from guanine to adenine (Supplementary Fig. 3b). To quantify the efficiency of all 12 possible nucleotide changes as well as the distance dependent efficiency decay, we edited the *FRDMD7* target with a DNA donor library containing 120 different donors that encode for all possible nucleotide changes at all 20 nucleotide positions left and right of the cut, together with HDRobust reagents to prevent competing indel formation[34] (Supplementary Fig. 3c, d). Thymine is preferentially exchanged with

cytosine, while an exchange into purines is disfavored (Supplementary Fig. 3e). In general, thymine and guanine are around twice as hard to substitute than adenine and cytosine (Supplementary Fig. 3f). The highest HDR efficiency can be achieved for substitutions directly at the cut site and an edit distance of just five nucleotides from the cut already halves HDR efficiency (Supplementary Fig. 3g).

## Prediction of synthetic gRNA activity

To achieve satisfactory prediction of synthetic Cas9 gRNA activity, we implemented the five features we identified to influence in vivo activity in a linear model. (Supplementary Data 2). Each of these features reduces gRNA activity and the combined cumulative detrimental features result in a score of potential gRNA activity (Fig. 3a). In style of the MIT off-target score named after the institution of its inventors we dubbed our in vivo synthetic gRNA activity score 'EVA activity score'. The EVA activity score highly correlates with our experimentally determined in vivo synthetic gRNA activity ($r = 0.83$) (Fig. 3b) and approaches the correlation we achieve between biological replicates ($r = 0.95$). An EVA score below 50 indicates inefficient gRNAs (all below the first quartile Q1, i.e., the worst 25% of gRNAs). To assess influence of gRNA delivery method and cell line on in vivo synthetic gRNA activity, we used a subset of nine gRNAs for editing of 409B2 hiPSCs, HEK293, and eHAP cells using Cas9 RNP electroporation. The EVA score shows similar correlations to in vivo synthetic gRNA activity independent of delivery and across tested cell lines ($r = 0.84–0.93$) (Fig. 3c). Then we validated the EVA score by determining the efficiency of forty gRNAs previously not used in model training. Importantly, we can predict the activity of those gRNAs with robust accuracy ($r = 0.74$) (Fig. 3d). To facilitate access to the EVA scores for the human genome, we precalculated the EVA scores for all gRNAs in the human and mouse genomes and provide online tracks for the USCS genome browser (Fig. 3e, see "Data availability" for links). Encouragingly, 95% of human genome-wide gRNAs are predicted to be efficient when chemically synthesized (Fig. 3f). Further, synthetic gRNAs are predicted to be able to edit 4105 of the 4140 disease-associated positions with a nearby a NGG PAM from a database for human variations and phenotypes (ClinVar)[35] in the human genome which are likely hard to edit with transcribed gRNAs, as they do not contain a single efficient nearby transcribed gRNA predicted by the Doench score[18] (Fig. 3g). As a follow-up study (Rule set 3)[27] showed that considering different tracrRNA architectures can lead to improved predictions for transcribed gRNAs, we also precalculated all Rule Set 3 gRNA predictions for the Hsu and Chen tracrRNA for the human genome for easy comparison in our genome browser track (Fig. 3e).

We also edited twenty different sites using gRNAs for Cas12a (Cpf1-Ultra)[36] (Supplementary Fig. 4a) and were surprised to find that, contrary to Cas9 tools, a prediction tool trained on transcribed Cas12a gRNA[37], can accurately predict in vivo synthetic Cas12a gRNA activity (DeepCpf1, $r = 0.77$) (Supplementary Fig. 4b). This may be due to the limited influence of the PAM-proximal sequence, typical of Cas12a (Supplementary Fig. 4c), as well as absence of the detrimental effect of low specificity on on-target cleavage efficiency we had observed with Cas9 (Supplementary Fig. 4d). While the available tools for Cpf1 trained on transcribed gRNAs can provide satisfactory prediction for synthetic Cas12a gRNA activity, they could still be refined for synthetic gRNA prediction by accounting for T-stretches that can terminate transcription but are not relevant for synthetic gRNAs (Supplementary Fig. 4e).

## Prediction of HDR efficiency

HDR efficiency is always below the predicted EVA score, but gRNAs with a high EVA score can still result in low HDR efficiency (Supplementary Fig. 5a). Due to the absence of available HDR efficiency prediction tools, we also generated a linear model (Supplementary Data 3) based on features we identified to influence HDR efficiency: gRNA

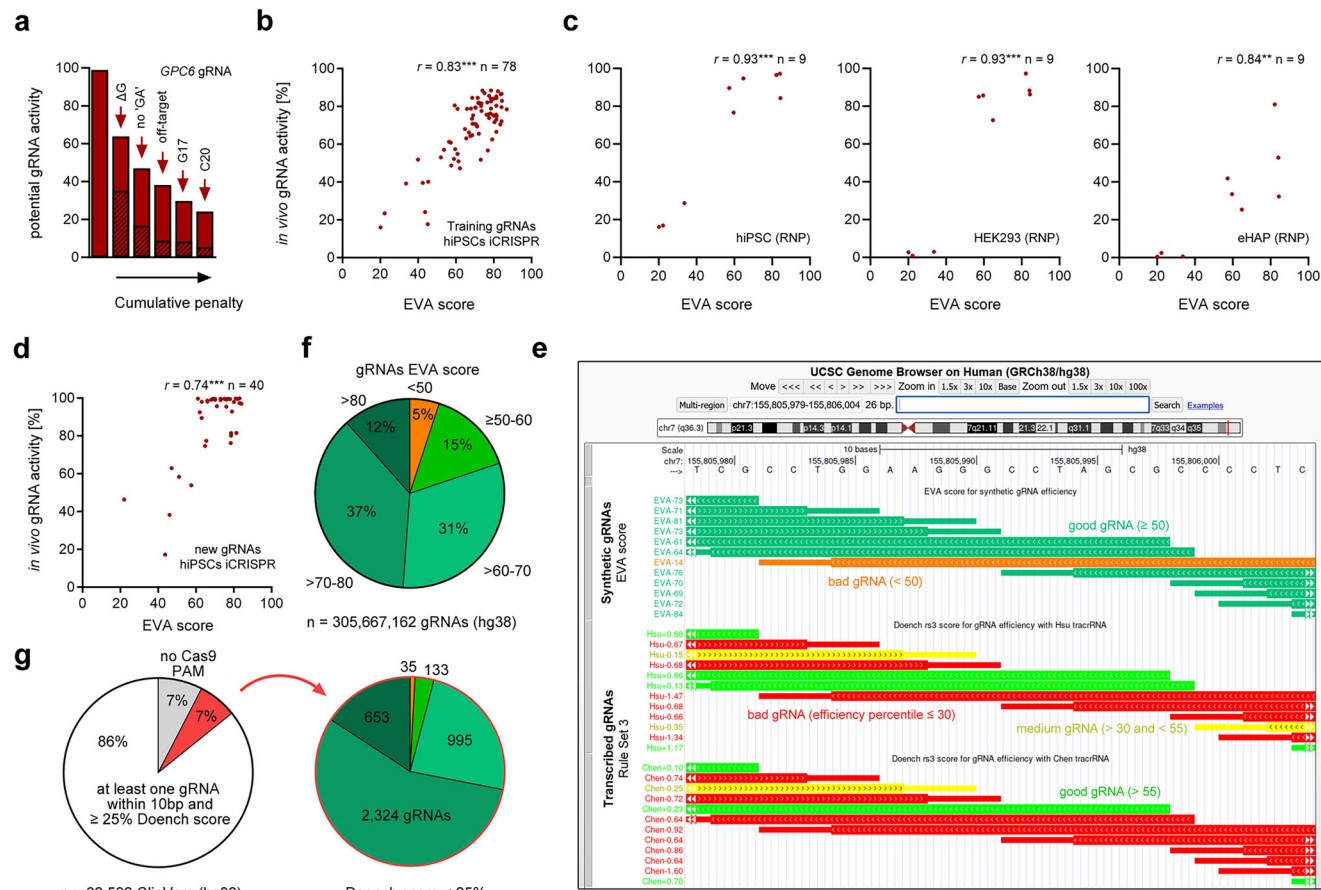

**Fig. 3 | Prediction of synthetic gRNA activity. a** Impact of detrimental gRNA features contributing to cumulative penalty on in vivo gRNA activity that constitutes the EVA activity score. The extent of penalty for the worst gRNA in the screen (*GPC6*) is shown. The shaded portion of the respective bars shows the individual feature penalty. **b** Scatter plot of the EVA activity score and measured in vivo gRNA activity from 409B2 iCRISPR hiPSCs used for training (*n* = 78). Each dot represents the mean of independent biological replicates (*n* = 3) for one gRNA. Pearson's r of the correlation (two-tailed) is stated (**\*\*p ≤ 0.01, \*\*\*p ≤ 0.001). c** EVA score and measured in vivo gRNA activity for RNP based editing (*n* = 9) in 409B2 hiPSCs, HEK293, and eHAP. Each dot represents the mean of independent biological replicates (*n* = 2) for one gRNA. **d** EVA score prediction performance for new gRNAs (*n* = 40) not used for training of the EVA score in 409B2 iCRISPR hiPSCs. Each dot represents the mean of independent biological replicates (*n* = 3) for one

gRNA. **e** Screenshot of the USCS Genome Browser[65] with a track for precalculated EVA scores that state the score and are color coded in green for efficient and orange for inefficient synthetic gRNAs. Below are gRNA tracks with Rule Set 3 scores for transcribed gRNAs with the Hsu or Chen tracrRNA[27], colored in percentile bins based on the already available Doench 2016 score CRISPR track[66]. **f** Percentage of predicted efficient synthetic gRNAs (EVA score bins ≥ 50) and inefficient synthetic gRNAs (EVA score < 50) for the human genome. **g** The left pie chart shows the percentage of human genome wide ClinVar[35] sites with at least one, zero due to missing proximal NGG Cas9 PAM, or no efficient proximal transcribed gRNA based on the Doench score. The right pie chart depicts the portion of ClinVar sites of the latter condition, that are predicted to be targetable when a synthetic gRNA is used instead. Source data are provided as a Source Data file.

activity, DNA donor free energy, and type of substitutions (Supplementary Fig. 3). This simple linear model highly correlates with of our experimentally determined HDR efficiency of the iCRISPR screen (*r* = 0.72) (Supplementary Fig. 5b), as well as for RNP based editing in 409B2, HEK293, and eHAP cells (*r* = 0.85–0.9) (Supplementary Fig. 5c). The HDR prediction score robustly predicts HDR efficiencies for targets on which the model was not trained (same sequence orientation as gRNA), including asymmetric donor designs[38] (reverse complement to gRNA), and different edit positions when multiplied with the distance efficiency decay factor from Supplementary Fig. 3g) (*r* = 0.59) (Supplementary Fig. 5d).

**Prediction performance of EVA score on published datasets**
In a next step, we tested the EVA scores performance on published Cas9 gRNA efficiency datasets (Supplementary Data 4). gRNA efficiencies from previous studies[18,32,39–44] tend to be lower than predicted by the ascribed EVA scores (Fig. 4a). This could be due to cellular repair outcomes that are not accounted for, as well as sequence dependent transcription bias of transcribed gRNAs, which both tend to

underestimate gRNA activity in these published datasets gRNA (Fig. 4b). If this is true, poor performing transcribed gRNAs with a good EVA score (>50) should be efficient when provided as synthetic gRNA. The four gRNAs with the lowest measured efficiency from the Doench et al. (2016) dataset containing a total of 2314 gRNAs[18] all have good EVA scores (range 70–87) and should thus be efficient when provided as synthetic gRNA (Fig. 4a last panel). To test this, we used these four worst and also the four best gRNAs from this dataset with Cas9 RNP editing of 409B2 hiPSCs and HEK293 with synthetic gRNAs. As anticipated, all eight tested synthetic gRNAs achieve efficient genome editing in both 409B2 hiPSCs and HEK293 cells (Fig. 4c, d). While the four worst transcribed gRNAs achieved 0–5% normalized phenotype depletion in the original study, the related synthetic gRNA achieve in vivo gRNA activities ranging from 79 to 96% (average 31-fold increase) and 57–97% (average 26-fold increase) in 409B2 hiPSCs and HEK293 cells, respectively. When directly comparing different gRNA architectures and delivery forms in 409B2 hiPSCs we observe comparable efficient in vivo activities for all eight targets for both synthetic hybridized crRNA:tracrRNA and synthetic sgRNA, while sgRNAs

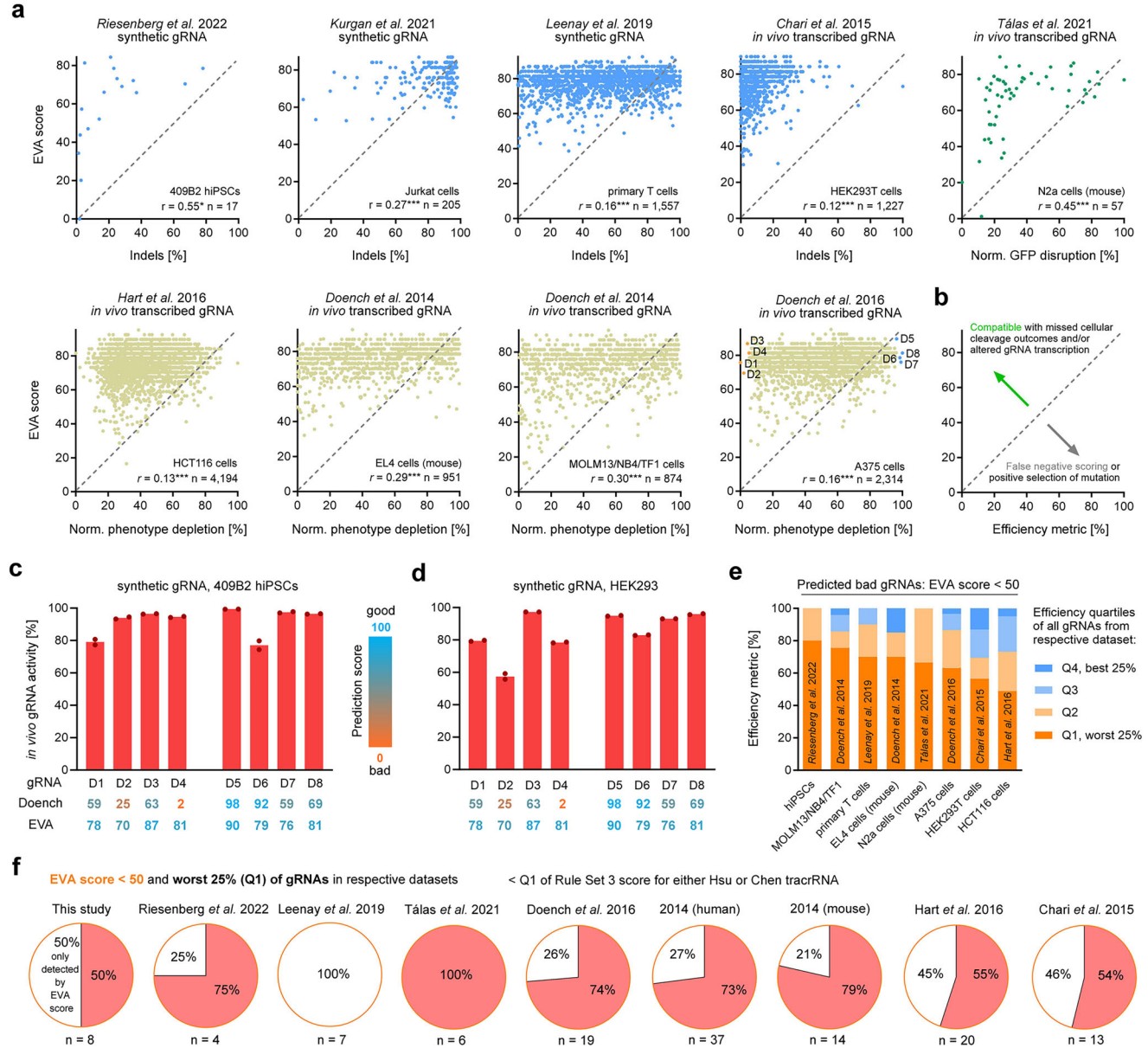

**Fig. 4 | EVA score prediction performance for published datasets. a** Scatter plots of gRNA efficiency metrics from published studies[18,32,39–44] and their respective EVA scores. Efficiency metrics are indels (blue), normalized GFP disruption (green), and normalized phenotype depletion (beige). Pearson's *r* (two-tailed) (***$p \leq 0.001$), cell type, and number of gRNAs in the datasets are stated. The four gRNAs with the lowest measured efficiency in the Doench et al. (2016) dataset are labeled orange (D1–D4), and the four gRNAs with the highest measured efficiency are labeled blue (D5–D8). **b** An EVA score higher than the measured efficiency (green arrow area) is compatible with missed cellular cleavage outcomes and/or gRNA transcription bias in the original studies, while EVA scores lower than the measured efficiency (gray arrow area) would tend to indicate false negative scoring by our prediction or positive selection of mutation. **c** In vivo gRNA activity of D1–D8 for editing using synthetic gRNA RNPs in 409B2 hiPSCs. Doench percentile scores (trained on the 2016 dataset including D1–D8) and EVA scores are stated and color coded for predicted gRNA efficiency. Independent biological replicates were performed (*n* = 2). **d** Same as **c** but in HEK293 cells. **e** Performance evaluation of EVA score prediction of bad gRNAs (EVA score < 50) in published datasets from (**a**). gRNAs are binned by quartiles of measured efficiencies. **f** Correctly predicted bad gRNAs (EVA score < 50) that belong to the worst 25% (Q1) of gRNAs of published datasets (see also Fig. 4e). The red portion of the pie chart describes the amount of those inefficient gRNAs, which would have also been predicted as inefficient by either the Hsu or the Chen tracrRNA Rule Set 3 score[27] (<Q1), and the white portion describes inefficient gRNAs that would have only been predicted as inefficient by the EVA score. Source data are provided as a Source Data file.

encoded in exogenous dsDNA and transcribed from a U6 promoter are less efficient for the previously described inefficient gRNAs from the Doench et al. (2016) dataset (Supplementary Fig. 6).

Consequently, a more meaningful way to benchmark our score would be assessing its ability to serve as an upper bound of the potential efficiency of a gRNA, i.e., if a gRNA has a bad EVA score (<50) it should also be inefficient irrespective of model system and gRNA production format used, and if a gRNA is efficient in other datasets it should have a good EVA score (>50). Indeed, an EVA score

below 50 robustly identifies inefficient gRNAs of studies that employed Cas9 genome editing in human cells (iPSCs, primary T cells, A375, MOLM, NB-4, TF1, HCT116, HEK293T)[18,32,40,42–44] and mouse cells (N2a, EL4)[41,42] with synthetic or transcribed gRNAs (Fig. 4e). Across datasets an average of 66% (range 49–80%) or 86% (range 70–100%) of gRNAs predicted to be inefficient were below Q1 or Q2 (i.e., the worst 50%) of the full datasets, respectively. The 25% percentile of the Rule Set 3 scores[27] for prediction of transcribed gRNAs can only identify a fraction of these inefficient gRNAs that

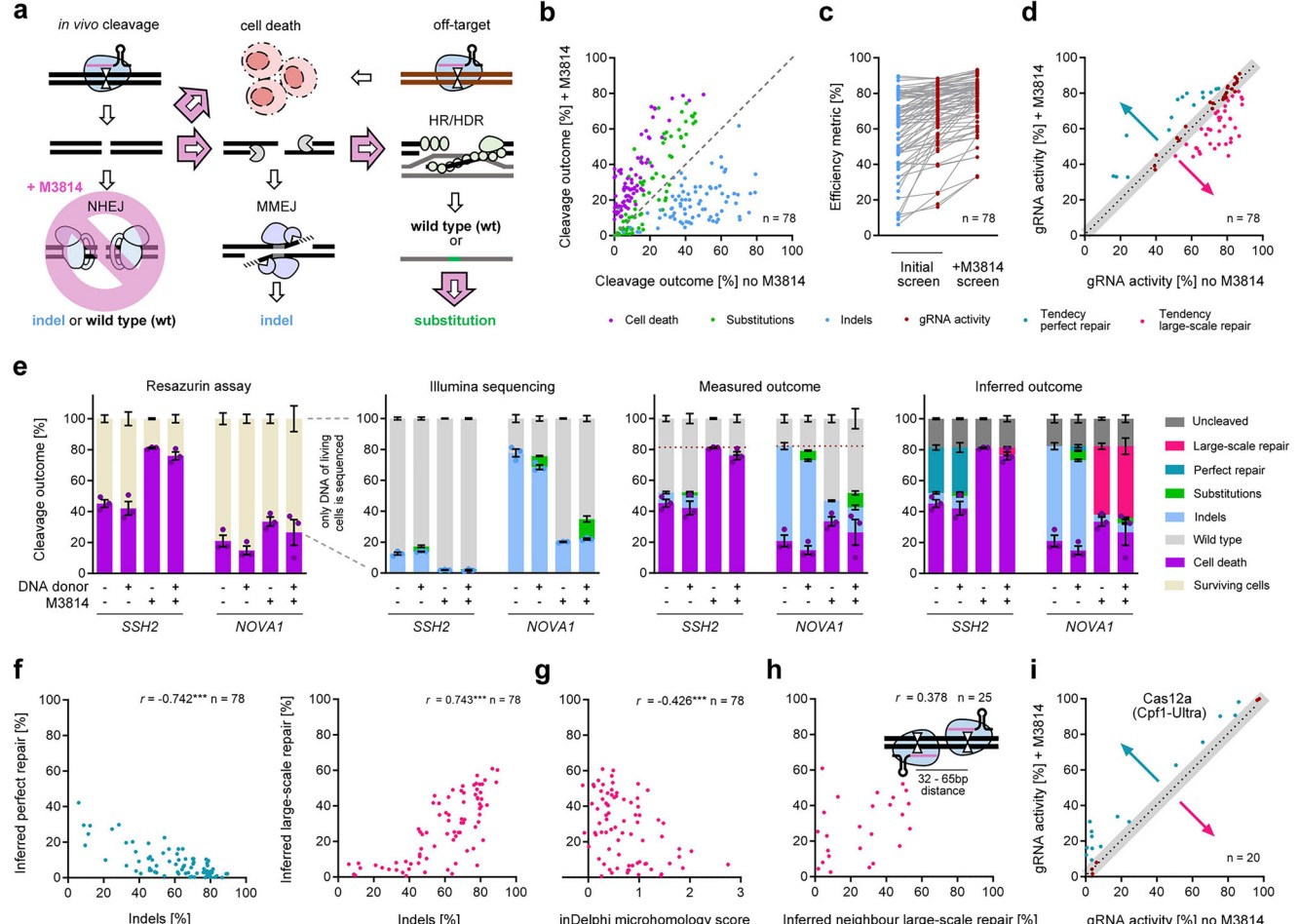

**Fig. 5 | Comparative screening with NHEJ inhibition to infer cryptic repair events. a** Schematic of potential cellular cleavage outcomes after a CRISPR induced DSB, when NHEJ is inhibited by M3814. **b** Scatter plot of the cleavage outcomes cell death, indels, and substitutions for editing with and without M3814. Each dot represents the mean of independent biological replicates ($n = 3$) for one gRNA ($n = 78$ targets). **c** Conventional indel metric and in vivo gRNA activity with and without M3814. **d** Scatter plot of in vivo gRNA activity with and without M3814. Because the gRNA activity is independent of repair pathway inhibition by M3814, deviation from a perfect correlation indicates either a tendency for perfect repair (turquoise dots), or large-scale repair (pink dots), both events that are cryptic and overlooked without comparative screening. The gray band indicates comparable activities ($\pm 5\%$) with or without M3814 treatment. **e** Resazurin assay, Illumina sequencing, and comparative screening with and without DNA donor or M3814 for the targets *SSH2* and *NOVA1*. The dotted lines between the resazurin assay panel and the Illumina sequencing panel indicate that only DNA extracted from surviving cells will be sequenced. The dotted line in the panel of measured outcomes marks the maximum detected cleavage efficiency that is used to infer the amount of perfect repair and large-scale repair Independent biological replicates were performed ($n = 3$) and error bars show the s.e.m. **f** Scatter plots of indels and inferred perfect repair (left panel) or large-scale repair (right panel). Pearson's $r$ (two-tailed) is stated (***$p \leq 0.001$). Each dot represents the mean of independent biological replicates ($n = 3$) for one gRNA ($n = 78$ targets). **g** Scatter plot of inDelphi[46] microhomology score and large-scale repair. **h** Scatter plot of inferred large-scale repair from two neighboring gRNAs (65 bp maximum cut distance). Twenty-five gRNAs have a neighboring tested gRNA in our screen. **i** Scatter plot of in vivo Cas12a gRNA activity with and without M3814. gRNAs with tendency for perfect repair are shown as turquoise dots. Each dot represents the mean of independent biological replicates ($n = 2$) for one Cas12a gRNA ($n = 20$ targets). Source data are provided as a Source Data file.

have measured data below Q1 of the respective datasets (Fig. 4f). Essentially no gRNA has an EVA score below 50, when assessing the top 25% of gRNAs based on measured data. (Supplementary Fig. 7). Thus, an EVA score below 50 alone appears to be sufficient to robustly separate inefficient from efficient synthetic gRNAs, and it will help to avoid inefficient transcribed gRNAs when combined with suitable prediction models for transcribed gRNAs.

**Comparative screening with NHEJ inhibition allows prediction of cryptic repair outcome tendencies**

Perfect DSB repair by either NHEJ or by HR with the sister chromatid is invisible to conventional targeted PCR. Since inhibition of NHEJ diverts repair to indel producing MMEJ and targeted substitution producing HDR[4] (Fig. 5a), we tested our initial set of gRNAs using DNA donors and NHEJ inhibition by M3814, a small molecule inhibitor of DNA protein kinase catalytic subunit[45]. M3814 decreases indels and cell survival for all targets, and increases targeted substitutions for most targets (Fig. 5b). NHEJ inhibition further increases the amount of measurable cellular gRNA cleavage for many targets. When data of the initial screen and the M3814 screen are combined, the quantified mean in vivo gRNA activity increases from 68% to 75% with a range of 33–93% (Fig. 5c).

Some targets appear to have higher quantified gRNA activity when NHEJ is inhibited, while others seem to be more efficient without NHEJ inhibition (Fig. 5d). Because CRISPR–Cas9 cleavage occurs prior to and thus uninfluenced of downstream DNA repair, the former targets arguably have a tendency for perfect repair by NHEJ which is not quantified without NHEJ inhibition, while the latter targets tend to be pushed into large scale repair by MMEJ[4] that cannot be quantified by PCR amplification. Inferred from our comparative screen, 19% of gRNAs tend to perfect repair, 49% have a tendency for large scale

repair, and the remaining 32% have comparable activities (±5%) with or without M3814 treatment. The comparative screening data allows us to infer the amount of cryptic perfect or large-scale repair for all tested gRNAs (Fig. 5e, Supplementary Data 5) (see Methods). Strikingly, perfect repair or large-scale repair is highly negatively ($r = -0.74$) or positively correlated ($r = 0.74$) to indels from the gRNA only screen (conventional metric), respectively (Fig. 5f). Further, a low microhomology strength score directly at the cut site[46] is predictive of a tendency for large-scale repair (Fig. 5g), while the neighboring genomic context is not (Fig. 5h). Surprisingly, 20 tested gRNAs for Cas12a all show a tendency for perfect repair, rather than large-scale repair, even though many of them have high activity (Fig. 5i).

To validate that our comparative screen can correctly infer a tendency for large-scale repair, we employed Cas9 RNP editing with gRNAs inferred to result in low or high frequency large-scale repair, generated single cell derived cellular clones, and performed a droplet digital PCR to quantify target locus copy number. Confirming correct inference, all 59 cellular clones from editing with two different gRNAs inferred to not have a tendency for large-scale repair have no copy number loss, while 11–28% of clones for two gRNAs with an inferred tendency for large-scale repair show copy number loss (Supplementary Fig. 8).

### Diverting perfect repair with paired gRNAs increases genome editing efficiency

In contrast to well described large-scale repair that can be experimentally validated by copy number detection[9–13], little is known about the extent of perfect repair efficiency for a DSB due to the inability to distinguish uncut and cut but perfectly repaired DNA sequences. To validate that our comparative screen can correctly infer a tendency for perfect repair, we set out to test if adjacent cleavage with an additional gRNA would remove the sequence in between the breaks and thus divert perfect blunt end repair (undetectable by PCR) to erroneous repair (detectable by PCR). We chose a first gRNA targeting SSH2 for which we had inferred frequent perfect repair (29%) and a second neighboring gRNA with high in vivo activity (71%). We used both gRNAs in combinations of Cas9 RNP nuclease, catalytically inactive dCas9, Cas9D10A nickase, and/or Cas9H840A nickase with a PAM-out gRNA pair (Fig. 6a) and quantified cell death and indel frequency (Fig. 6b).

Similar to our screen, the gRNAs result in low indels (12%) or high indels (48%) when used alone with Cas9 for the first or the second gRNA, respectively. While additional binding of dCas9 or nicking with Cas9H840A did not increase indel frequency compared to use of the first gRNA with Cas9, additional nicking with Cas9D10A increased indel frequency to 43% (3.6-fold) and thus achieved comparable efficiency to Cas9 editing when using the second gRNA alone. The strictly vertical flank of the peak at the first gRNA break site in the indel pattern with an additional gRNA of the latter conditions suggests efficient cleavage of the low indel inducing gRNA (Fig. 6c). Thus, decent cleavage of the first gRNA is indeed masked by precise repair of the blunt Cas9 cut to the wild type state that cannot be distinguished from uncleaved wild-type cells. Encouragingly, this method using additional nicking also suggest 29% perfect repair for the first SSH2 gRNA (Fig. 6b, top panel), which is identical to the amount of inferred perfect repair from the comparative screen (Fig. 5e)

To further confirm additional gRNA nicking as an approach to increase indel frequency, we also compared single gRNA Cas9 nuclease RNP editing with paired gRNA Cas9 nuclease/D10A-nickase RNP editing for four more targets. Additional nicking increased indels across all those targets from 0.7–28% to 15–77% (mean 9.6-fold increase) (Fig. 6d). Deletion shapes for additional nicking show clear vertical flanks at the DSBs, and these flanks are not present in absence of additional nicking (Fig. 6e). The near perfect correlation ($r = 0.99$) to previously determined in vivo gRNA activity shows that additional nicking can channel essentially all gRNA activity into 'productive'

genome editing events like indels (Fig. 6f). As NHEJ inhibition by M3814 also prevents perfect repair at DSBs, it can also increase erroneous repair without the need of additional nicking. We show that this is especially powerful when combined with an improved gRNA backbone engineered for better efficiency (GOLD-gRNA)[32] (Fig. 6g). For RNP editing using a high-fidelity Cas9 variant (HiFi)[47] a combination of M3814 and GOLD-gRNA increased indels across four targets from 2.6–42% to 74–94% (mean 10.6-fold increase). Although both the SOD1 and the OSBP2 gRNA have the worst possible Doench 2016 score (0), application of the synthetic GOLD-gRNA with M3814 achieves 74% and 82% genome editing, respectively.

## Discussion

Chemically synthesized gRNAs are widely used in combination with highly efficient RNP delivery[34,39,40,47,48] and could be the gRNA type of choice for many CRISPR-related medical applications, as chemical modifications can be introduced that are present in virtually all clinically approved RNAs[49]. Contrary to transcribed gRNAs, synthetic gRNAs are not restricted to start with a guanine or less favored adenine[7] and are not influenced by sequence dependent transcription yield or bias. For future gene therapy, DNA-free gene editing using synthetic gRNA and Cas9 mRNA/protein would seem preferable to avoid integration of plasmids used to produce gRNAs into the genome. Given that one third of CRISPR researchers use synthetic gRNAs[48], there is a need for a robust prediction tool for synthetic gRNA activity.

Numerous studies have investigated features that can influence activity of transcribed gRNAs[6–8,18,26,31]. Even though they sometimes predict activity in their own datasets well, the correlation between predicted and actual efficiency for other datasets generated with transcribed gRNAs varies considerably[17] and is unsatisfactory for our dataset generated with synthetic gRNAs (Fig. 2a). We attribute the inability of existing tools to predict indels or gRNA activity for synthetic gRNAs to two main factors. First, these existing tools are likely influenced by sequence dependent transcription bias of gRNAs, 5′- and 3′- inhomogeneity[19–22], and premature transcription termination[23]. Second, we do not only sequence indels introduced by NHEJ and MMEJ at the target site (previously done for eukaryotic screens[7,8]), but also quantify targeted substitutions introduced by HDR, and measure cell survival (previously done for prokaryotic screens[41]) in hiPSCs with diploid copy number (Fig. 1a, b). Consequently, our in vivo gRNA activity metric considering indels, targeted substitutions, and cell death captures a more accurate representation of cellular cleavage outcomes, and is thus an optimized metric for cellular gRNA cleavage efficiency. This is supported by our observation that in vivo gRNA activity is correlated with in vitro gRNA cleavage, but not with indels (Supplementary Fig. 2e). However, in vitro gRNA cleavage can strongly differ from in vivo gRNA cleavage and is thus not suitable for in vivo gRNA activity prediction. This could be due to many factors such as cellular environment vs. synthetic buffer, and complex genome vs. short oligos.

Strikingly, 93% of gRNAs from our screen have at least around 40% in vivo gRNA activity and the rare inefficient gRNAs have only few features responsible for their inefficiency (Fig. 2c): very negative free energy of the 20nt gRNA spacer (gRNA misfolding[7,50]), unfavored cut site nucleotides (potentially nuclease disfavored bases or ligation/1 bp integration bias[8]), PAM-proximal C20 (extended efficient PAM requirement $N_{19}DNGG$), low number of 'GA' dinucleotides (might influence gRNA stability via stacking interactions[51]) low MIT off-target score (sequestering of gRNA to off-target sites[52,53]). Our observation of an impact of gRNA specificity on in vivo gRNA activity, which had not been tested in previous gRNA efficiency prediction tools, is in line with the ability of DeepHF to achieve a better prediction when Cas9 high fidelity variants were used compared to wild type Cas9[7], presumably since confounding off-target binding was reduced in the former. Contrary to transcribed gRNAs, synthetic gRNAs show a strongly

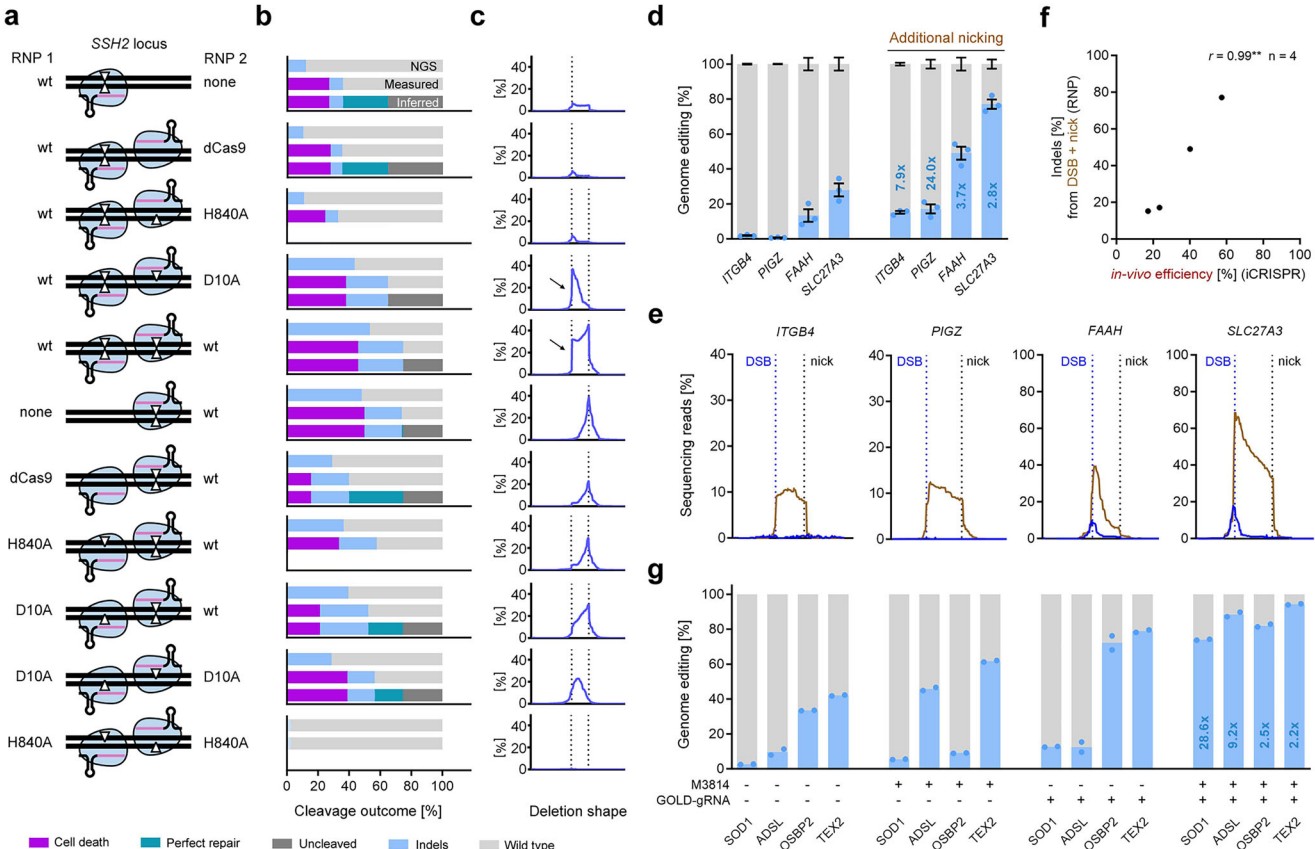

**Fig. 6 | Diverting perfect repair increases editing efficiency. a** Schematic tested combinations of no enzyme, wild type Cas9 (wt), dead Cas9 (dCas9), Cas9-D10A, and Cas9-H840A for editing of the *SSH2* locus using two neighboring PAM-out gRNAs. The triangles indicate the active nuclease domains. **b** Illumina sequencing, measured outcomes (combined cell death and sequencing), and inferred perfect repair for RNP-based editing in 409B2 hiPSCs for combinations from a). Independent biological replicates were performed (*n* = 2). Due to the lower cleavage efficiency of the H840A variant, outcomes were not inferred for combinations with H840A. **c** Deletion frequency percentage and deletion shapes from Illumina sequencing for combinations from (**a**). Dotted lines indicate the positions of gRNA cleavage sites. Vertical flanks of deletion shapes at cleavage sites are shown by arrows. **d** Genome editing efficiencies for four targets with a single gRNA (wild type Cas9 RNP), and with an additional neighboring nicking gRNA (Cas9–D10A RNP). Fold-changes of editing efficiency are stated. Independent biological replicates

were performed (*n* = 3) and error bars show the s.e.m. **e** Deletion frequency percentage and shapes from Illumina sequencing from (**d**). The blue lines indicate the deletion shape of editing using a single wild type Cas9 gRNA, and the blue dotted lines indicates the cleavage site of said gRNAs. The brown lines indicate the deletion shape with an additional neighboring nicking gRNA, and the black dotted line indicates the additional nicking site. **f** Scatter dot plot of indels from (**d**) with additional nicking and in vivo gRNA efficiency determined in the initial synthetic gRNA screen. Pearson's *r* (two-tailed) is stated (**$p \leq 0.01$). **g** Genome editing efficiencies for four targets with a Cas9-HiFi RNP in 409B2 hiPSCs with or without M3814 and normal gRNA backbone or GOLD-gRNA[32], as well as combination of M3814 and GOLD-gRNA. Fold-changes of editing efficiency are stated. Independent biological replicates were performed (*n* = 2). Source data are provided as a Source Data file.

reduced impact of PAM-proximal bases on activity (Fig. 2b), albeit more data is needed to accurately quantify their impact in synthetic gRNAs. This difference might be due to transcription bias and/or increased stability of synthetic gRNAs, as increased expression of gRNAs has been shown to reduce impact of PAM-proximal bases[53]. This makes synthetic gRNAs a potent alternative for transcribed gRNAs with suboptimal PAM-proximal nucleotides, like those with PAM-proximal dinucleotides comprised of adenines and/or thymines, which make up a quarter of possible gRNAs.

Identification of detrimental features allowed us to develop a simple hypothesis-driven linear model, the EVA activity score, that (I) achieves reliable prediction of in vivo Cas9 synthetic gRNA activity (Pearson *r* = 0.74) (Fig. 3d), (II) can serve as an upper bound of the potential efficiency of a gRNA, regardless of whether it is produced by transcription or synthesis (Fig. 4a), and (III) can robustly identify bad gRNAs (EVA score < 50) for our own experiments, as well as for several published datasets (Fig. 4e). However, more in vivo gRNA activity data from additional gRNAs could improve future models, as the current number used to train our model may limit generalizability. The EVA

score is often much higher than the measured efficiencies in published datasets, which does not seem to be due to poor performance of the score, but rather to better performance of gRNAs when delivered in chemically synthesized form, as the four worst published transcribed gRNAs from the largest dataset all have more than 79% synthetic gRNA activity (Fig. 4c, d, Supplementary Fig. 6). Even though the synthetic gRNA dataset from Leenay et al.[40] is free of gRNA transcription bias, EVA scores are still often much higher than the measured indels (Fig. 4a). While this could be due to false positives, which would reflect a limitation of the EVA score, few measured indels and a high EVA score would also be compatible with high gRNA cleavage induced death of the primary human T cells used in their study, which are uniquely susceptible to DNA damage[54].

For Cas12a, DeepCpf1[37] achieves reliable prediction of in vivo Cas12a synthetic gRNA activity even though it has been trained on transcribed gRNAs (Supplementary Fig. 4). Cas12a has a shorter gRNA, cleaves double-stranded DNA further away and 3' from the PAM via staggered cuts that generated sticky ends[55], and has a higher specificity[56], all features which likely contribute to more

robust prediction of Cas12a gRNA efficiency compared to Cas9 gRNA efficiency.

In addition to prediction tools for gRNA efficiency, several tools to predict indel pattern outcomes have been developed[40,46], but no tool for predicting HDR efficiency is available, despite the widespread use of HDR-dependent editing for point mutations. Our findings that HDR efficiency can be limited by DNA donor misfolding due to very negative free energy and suboptimal efficiency for some types of nucleotide changes (Supplementary Fig. 3a, b) allowed us to create a simple linear model for HDR prediction. Interestingly, thymine and guanine tend to be changed less efficiently than adenine and cytosine for HDR, and the same has been described for prime editing as well[57]. Excitingly, when DNA donor sequence characteristics are optimal, an exceptional HDR efficiency of up to 78% can be achieved (Supplementary Fig. 5d).

As for DNA donor sequence features influencing HDR efficiency, not much was known about impact of gRNA sequence on large-scale or perfect DSB repair, which both can constitute more than a third of repair outcomes[12–15,58]. Both repair events are 'cryptic', i.e., they cannot be detected by standard target PCR due to lack of primer sites for the former and indistinguishability from the uncut wild type sequence for the latter. Current methods to detect such unintended effects are cumbersome. Large scale repair quantification relies on generation of single cell derived cellular clones[12,14], and quantification of perfect repair of DSBs had been attempted by mathematical modeling combined with time-course measurements of indel accumulation[58]. Advancing these approaches, our comparative screen of gRNAs with and without NHEJ inhibition can infer tendency of perfect repair and large-scale repair at high-throughput (Fig. 5). Of note, this comparative screen approach is limited in that it cannot infer large scale repair when NHEJ is not inhibited, but is rather amplifying a tendency of a target to undergo large-scale repair when NHEJ is inhibited[59]. We show that both high Cas9 gRNA activity and low microhomology strength directly at the cut site are predictive of a strong tendency for large-scale repair (Fig. 5f, g), which is in line with previous evidence that size of deletions increases with the overall efficiency of Cas9 mutagenesis[59]. In contrast, Cas9 gRNAs that result in few indels tend to produce DSBs that are largely repaired perfectly.

Strikingly, when editing with Cas12a, Cas12a gRNA cryptic cleavage outcomes tend to undergo perfect repair rather than large-scale repair, even when having high activity (Fig. 5i). It is tempting to speculate that sticky ends generated for Cas12a compared to blunt ends generated by Cas9 influence DSB repair sub-pathways[24,60] resulting in a drastically reduced tendency for large-scale repair for Cas12a. In line with this, it has been recently shown that Cas12a can reduce translocations compared to use of Cas9[61]. Since both unintended translocations and chromosome loss have been described after editing of T cells from patients with Cas9[62–64], prevention of cryptic unintended large-scale repair including chromosome loss could be a major advantage of Cas12a over Cas9 for safety in gene therapy applications.

## Methods

### In vitro cleavage assay
For in vitro cleavage of target DNA, a mix consisting of 340 nM gRNA (hybridized crRNA:tracrRNA) and 165 nM Cas9 (Integrated DNA Technologies) was incubated for 20 min at 20 °C, and subsequently mixed with 75 nM of the corresponding double-stranded DNA targets (see Supplementary Data 6 for oligonucleotides used in this study), that had been previously generated by hybridizing complementary single-stranded DNA for 2 min at 95 °C followed by 10 min at 20 °C. Samples were incubated at 37 °C for 16 h and then subjected to Proteinase K (NEB, catalog no. P8107S, 800U/ml) and RNaseA (Qiagen, catalog No. 1007885, 2 mg/ml) treatment. Proteinase digest was carried out at 50 °C for 30 min before RNaseA digest at 37 °C for 30 min. Samples were subjected to to capillary gel electrophoresis with a Fragment Analyzer (Advanced Analytical).

### Cell culture
For this study we used the previously described iCRISPR-Cas9 line from 409B2 hiPSCs[24] (GMO permit AZ 54-8452/26). Cells were grown on Matrigel Matrix (Corning, catalog No. 35248) in mTeSR1 medium (StemCell Technologies, catalog No. 05851) with supplement (StemCell Technologies, catalog No. 05852) and MycoZap Plus-CL (Lonza, catalog no. VZA-2011) that was replaced daily. HEK293 cells (Cytion, catalog No. 300192) were grown in Dulbecco's modified Eagle's medium/F-12 (Gibco, catalog no. 31330-038) supplemented with 10% fetal bovine serum (FBS) (SIGMA, catalog no. F2442), MycoZap Plus-CL, and 1% NEAA (SIGMA, catalog no. M7145). Haploid eHAP cells (Horizon, catalog No. C669) were grown in Iscove's modified Dulbecco's media (ThermoFisher, catalog No. 12440053) with 10% FBS and MycoZap Plus-CL. At ~80% confluency, cells were dissociated using EDTA (VWR, catalog No. 437012C) and split 1:6 to 1:10. For hiPSCs the medium was further supplemented with 10 µM Rho-associated protein kinase (ROCK) inhibitor Y-27632 (Calbiochem, catalog No. 688000) for 1 day after replating. Cells were grown at 37 °C in a humidified incubator with 5% $CO_2$. For generation of single cell derived cellular colonies (Supplementary Fig. 8), cells were dissociated using TrypLE Express (ThermoFisher, catalog No. 12605036) and triturated, followed by sorting of single cells in wells of a 96-well plate using a single-cell printer (Cytena).

All cell lines were authenticated by the supplier via certificate of analysis and additionally in-house by checking morphology. All cell lines were tested negative for mycoplasma contamination before and after the experiments.

### Lipofection of oligonucleotides
409B2 iCRISPR–Cas9 hiPSCs were incubated in medium containing 2 µg/ml doxycycline (Clontech, catalog no. 631311) 3 d before lipofection to express Cas9. Lipofection of oligonucleotides was done with a final concentration of 15 nM of each gRNA that had been previously formed by hybridization of crRNA and tracR, and 10 nM of single-stranded DNA donor or non-homologous DNA. The DNA donors have gRNA spacer sequence orientation, carry a transition mutation (A:G or T:C) at the cut site, and have homology arms of 45nt (Supplementary Data 6). For the screen, we used gRNAs from previous and planned evolutionary studies, mainly selected to cut close to genomic sites where modern humans and archaic humans differ. 0.75 µl RNAiMAX (Invitrogen, catalog No. 13778075) and the respective oligonucleotides were separately diluted in 25 µl OPTI-MEM (Gibco, catalog No. 1985-062) each and incubated at 20 °C for 5 min. Both dilutions were mixed to yield 50 µl of OPTI-MEM including RNAiMAX and oligonucleotides. The lipofection mix was incubated for 20–30 min at 20 °C. During incubation cells were dissociated using EDTA for 5 min and counted using the Countess Automated Cell Counter (Invitrogen). The lipofection mix, 100 µl containing 25,000 dissociated cells in mTeSR1 supplemented with Y-27632 and 2 µg/ml doxycycline were mixed and put in 1 well of a 96-well plate covered with Matrigel Matrix (Corning, catalog No. 35248). After 24 h, the medium was replaced with normal mTeSR1. Where applicable, the medium contained 2 µM M3814 for 2 days before switching to normal mTesR1.

### Electroporation of oligonucleotides and RNPs
Cells were treated with TrypLE (Gibco, catalog No. 12605010) for 5 min at 37 °C and triturated to obtain single cells, before addition of pre-heated media. Cells were counted using the Countess Automated Cell Counter (Invitrogen) and cell suspensions were centrifuged at 300g for 3 min. For cells without integrated iCRISPR, we used recombinant *Streptococcus pyogenes* Cas9, Cas9D10A, Cas9H9840A, dCas9, Cas9-HiFi (R691A) proteins, as well as the *Acidaminococcus sp.* BV3L6 Cas12a (Cpf1-Ultra) protein from Integrated DNA Technologies. Electroporation for all cell types was done using the B-16 program of the

Nucleofector 2b device (Lonza) in cuvettes for 100 μl Human Stem Cell nucleofection buffer (Lonza, catalog no. VVPH-5022), containing 1 million cells, 100 pmol electroporation enhancer, 320 pmol gRNA (crRNA/tracR duplex for Cas9 and its variants and crRNA for Cas12a), 252 pmol CRISPR enzyme, and 200 pmol of single-stranded DNA donor.

## Illumina library preparation and sequencing

At least days after transfection cells were detached using TrypLE (ThermoFisher, catalog No. 12605036), pelleted, and resuspended in 15 μl QuickExtract (Lucigen, catalog No. QE0905T). Incubation at 65 °C for 10 min, 68 °C for 5 min and 98 °C for 5 min was performed to yield single stranded DNA as a PCR template. Primers for each targeted loci were designed to contain adapter overhangs for Illumina sequencing (Supplementary Data 6). PCR was done in a T100 Thermal Cycler (Bio-Rad) using the KAPA2G Robust PCR Kit (SIGMA, catalog No. KK5024) with buffer B and 3 μl of cell extract in a total volume of 25 μl. The thermal cycling profile of the PCR was: 95 °C 3 min; 34× (95° 15 s, 65 °C 15 s, 72 °C 15 s); 72 °C 60 s. Illumina adapters with sample specific indices were added in a subsequent PCR reaction[65] using Phusion HF MasterMix (Thermo Scientific, catalog No. F-531L) and 0.3 μl of the first PCR product. The thermal cycling profile of the second PCR was: 98 °C 30 s; 25× (98° 10 s, 58 °C 10 s, 72 °C 20 s); 72 °C 5 min. Amplifications were verified by size separating agarose gel electrophoresis using 2% EX gels (Invitrogen, catalog No. G4010-11). The indexed amplicons were purified using Solid Phase Reversible Immobilization (SPRI) beads[66]. Double-indexed libraries were sequenced on a MiSeq (Illumina) giving paired-end sequences of 2 × 150 bp (+7 bp index). After base calling using Bustard (Illumina) adapters were trimmed using leeHom[67].

## Droplet digital and quantitative PCR

Copy numbers of target sequences were estimated by quantitative ddPCR. Primers were designed flanking the cut site and the probe was designed excluding edited sites sequencing (Supplementary Data 6). The gene *FOXP2* was used as copy number reference. The ddPCR amplification was done in × ddPCR Supermix for probes (no dUTP, Bio-Rad, catalog no. 1863024), 0.2 μM primer and 0.2 μM probe for target and reference, together with 1 μl genomic DNA in QuickExtract DNA extraction solution. After droplet generation, the PCR reaction was run for 5 min at 95 °C, followed by 42 cycles of 35 s at 95 °C (at a ramp rate of 1.5/2 °C s⁻¹) and 65 s at 60 °C (at a ramp rate of 1.5/2 °C s⁻¹) and 5 min at 98 °C. Droplets were read in a QX200 Droplet reader (Bio-Rad) and allele copy numbers were determined relative to a different fluorophore for the *FOXP2* reference and unedited control.

## Amplicon sequence analysis

Bam-files were demultiplexed and converted into fastq files using SAMtools (v1.12)[68]. CRISPResso (v1)[69] was used to analyze fastq files for percentage of wild type (unedited), targeted nucleotide substitution (HDR), and indels (NHEJ and MMEJ). Analysis was restricted to amplicons with a minimum of 70% similarity to the wildtype sequence and to a window of 20 bp from each gRNA. Sequence similarity for an HDR occurrence was set to 95%. Unexpected substitutions were ignored as putative sequencing errors. Due to absence of an annotated gRNA efficiency table in the original study of Kurgan et al., we downloaded their sequencing data from the SRA under accession number SRA:PRJNA675792 and performed CRISPResso analysis to obtain indel frequencies. Sequencing data from single cell-derived cellular clones was analyzed using SAMtools.

## Resazurin assay

Three days after transfection of editing reagents, cells were supplied with media containing 10% resazurin solution (Cell Signaling, catalog No. 11884) and grown for 5 h before fluorescence readings using a Typhoon 9410 imager (Amershamn Biosciences) and quantification using ImageJ and the 'ReadPlate' plugin (Figs. 1, 3d, 5, and 6), or using a CLARIOstar imager (BMG Labtech) (Figs. 3c, 4, and Supplementary Fig. 5). Resazurin is converted into fluorescent resorufin by cellular dehydrogenases and fluorescence (excitation: 530–570 nm, emission: 590–620 nm) reflects the amount of living cells[25]. Wells with media and resazurin but without cells were used as blank. Resazurin fluorescence for lipofection with a gRNA designed not to have a target in the human genome was considered as 100% cell survival.

## Linear models for prediction of synthetic gRNA activity and HDR efficiency

Free energy calculation was done using the 'oligoscreen' function of RNAstructure web server[70]. MIT off-target scores[33] calculated by the CRISPOR tool[71,72] were extracted from the USCS genome browser 'CRISPR Targets Track'[73,74]. The EVA score model was fitted using linear regression as implemented in the R function lm using R version 4.2.2 after transforming data of input of free energy (≤ −3 with $r = 0.74$ vs. full $\Delta G$ range of with $r = 0.59$) and MIT specify score (≥75 with $r = 0.37$ vs. full MIT range with $r = 0.34$) to account for non-linearity of in vivo gRNA efficiency when larger than these boundaries. The parameters of the linear model and an EVA score calculator excel sheet is given in Supplementary Data 2. If the free energy and/or MIT score of a gRNA is bigger than −3 and/or 75, the input needs to be −3 and/or 75, respectively. The HDR score model was fitted using linear regression as implemented in the R function lm using R version 4.2.2 with input of EVA score, DNA donor free energy, and nucleotide change penalty. The parameters of the linear model and an HDR score calculator excel sheet is given in Supplementary Data 3.

## Genome-wide EVA score calculation

To calculate EVA score for all possible gRNAs in the genome, first their sequences and corresponding MIT specificity scores were extracted from USCS genome browser table 'crispr.bb'. Free energy calculation was done using the 'oligoscreen' function in RNAstructure[70] and EVA score was calculated. Rule Set 3 scores for both the Hsu and Chen gRNA backbones were calculated using the original authors' Python package[27]. Those scores were provided as bigBed file and can easily be accessed in USCS genome browser via the following links: hg38 (human) (https://genome-euro.ucsc.edu/cgi-bin/hgTracks?db=hg38&hubUrl=https://bioinf.eva.mpg.de/pub/EVA_USCS_tracks/hub.txt) and mm39 (mouse) (https://genome-euro.ucsc.edu/cgi-bin/hgTracks?db=mm39&hubUrl=https://bioinf.eva.mpg.de/pub/EVA_USCS_tracks/hub.txt).

## ClinVar variant nearby cleavage calculation

ClinVar variants file was downloaded (release 2023-04-16) and it contained 4,453,543 entries of which 294,092 were flagged as pathogenic. These were further filtered for sites that are single nucleotide variants (SNVs), are not on chrMT, and that have both the reference and alternative allele reported for the GRCh38 genome. Of the filtered 62,492 pathogenic SNVs, 53,707 can be cut within 10 bases by a gRNA with Doench score of 25% or greater. Of the remaining 8885 sites for which there is no gRNA with a high Doench score, 4105 have a cut site within 10 base distance for gRNA with an EVA score of 50 or higher, 35 have an EVA score that is lower than 50 and 4618 have no cut site at all.

## Inference of editing outcomes

Inference of editing outcomes is done to recapitulate the entirety of cellular editing outcomes including those that result in cell death. The resazurin assay and Illumina NGS sequencing are measures of cell survival and editing, respectively. Since NGS editing efficiencies are only quantified from surviving cells, the amount of cell death can be

used to relate NGS editing efficiencies to their portion in the initial cell population directly after cleavage.

$$\text{death corrected NGS frequency} = (100 - \text{cell death}) \times \left(\frac{\text{NGS frequency}}{100}\right) \tag{1}$$

The gRNA activity metric combines both cell death and NGS editing:

$$\text{gRNA activity} = \text{cell death} + (100 - \text{cell death}) \times \left(\frac{\text{NGS frequency}}{100}\right) \tag{2}$$

Calculation of the apparent gRNA activity from all conditions of the comparative screen (with or without donor and/or M3814) provides the maximum gRNA activity of all conditions. This maximum gRNA activity represents the cleavage efficiency closest to the ground truth and can be used to infer refer 'cryptic' repair events when metric deviate from this ground truth. On the one hand, when the measured gRNA activity is lower for conditions without M3814, the difference of quantified cleavage to the maximum gRNA activity is considered the amount of perfect repair. On the other hand, when the measured gRNA activity is higher for conditions without M3814, the difference of quantified cleavage to the maximum gRNA activity can be used to infer the amount of large-scale repair. In this case, it is necessary to further adjust corrected NGS frequencies. Here, NGS editing efficiencies are only quantified from the subset of surviving cells without large-scale repair events, the amount of both cell death as well as large-scale repair events are thus necessary to relate NGS editing efficiencies to their portion in the initial cell population directly after cleavage.

$$\text{large scale repair correction factor for NGS frequency}$$
$$= \frac{(100 - \text{maximum gRNA activity})}{(100 - \text{gRNA activity})} \tag{3}$$

### Statistics and reproducibility

Bar graphs in figures were plotted, and s.e.m. error bars and correlations were calculated using GraphPad Prism (v6) software. The number of replicates is stated in the respective figure legends. No statistical method was used to predetermine sample size. The experiments were not randomized. Samples were prepared unblinded but in parallel. Analysis was performed on the basis of numerical sample names, without the identity of the samples being known during the analysis.

### Reporting summary

Further information on research design is available in the Nature Portfolio Reporting Summary linked to this article.

## Data availability

The sequencing data generated in this study are deposited in the NCBI's Sequence Read Archive (SRA) with the accession code PRJNA1258195 and are available on request from the authors. Source data are provided with this paper. gRNA scores in the USCS genome browser can be assessed via the following links: hg38 (human) (https://genome-euro.ucsc.edu/cgi-bin/hgTracks?db=hg38&hubUrl=https://bioinf.eva.mpg.de/pub/EVA_USCS_tracks/hub.txt) and mm39 (mouse) (https://genome-euro.ucsc.edu/cgi-bin/hgTracks?db=mm39&hubUrl=https://bioinf.eva.mpg.de/pub/EVA_USCS_tracks/hub.txt).

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

## Acknowledgements

We thank S. Pääbo for support and constructive input, H. Zeberg for help with the linear model, and A. Weihmann and B. Schellbach for DNA sequencing. We further thank E. Moreb for providing a table of the averaged activity for all PAM-proximal 5-mers from the study of Moreb and Lynch 2021. Funding was provided by the Max Planck Society (S. Pääbo) and the NOMIS foundation (S. Pääbo).

## Author contributions

S.R. conceived the idea and designed the study. S.R., P.K. performed editing experiments. S.R., P.K., T.M., and R.K. analyzed data. S.R., P.K., and T.M. gave input on study design. S.R. and T.M. wrote the paper with input from all authors.

## Funding

## Competing interests

Related patent applications on DNA-PKcs inhibitors for increasing genome editing efficiency (patent applicant: Max Planck Society; inventors: S.R. and T.M.; application number: EP18215071.4) and GOLD-gRNA (patent applicant: Max Planck Society, inventors: S.R., N.H., and T.M. application number: EP21176366.9) have been filed. The remaining authors declare no competing interests.
