## [Transparent Peer Review file · Nature Communications]

Robust prediction of synthetic gRNA activity and cryptic DNA repair by disentangling cellular CRISPR cleavage outcomes

Corresponding Author: Dr Stephan Riesenberg

Version 0:

Reviewer comments:

Reviewer #1

(Remarks to the Author)

In this study, the authors presented three key results relevant to the prediction of synthetic Cas9 guide RNAs (gRNAs) activity. First, they screened 78 synthetic gRNAs with or without corresponding single-stranded DNA donors to detect evidence of gRNA activity from various DNA repair mechanisms. They further investigated the effect of including non-homologous end joining (NHEJ) inhibition using this set of gRNAs to infer the tendencies of perfect repair and large-scale repair. Second, they employed these data to assess features previously found to be associated with transcribed gRNA activity and develop the EVA activity score, a linear model with five features to predict synthetic gRNA activity. Following the model's development, the authors assessed the score on published datasets. Third, in response to EVA score overestimating gRNA activity when detected by homology directed repair (HDR), they developed a linear model for HDR efficiency prediction. The authors additionally reported that, despite suboptimal performance of models trained on transcribed Cas9 gRNA data to predict synthetic gRNA activity, models trained on transcribed Cas12a gRNA data yield more reliable predictions in the synthetic context, and identify direction for optimization. Moreover, they demonstrate that editing with Cas12a exhibits a stronger tendency towards perfect repair over large-scale repair as compared to Cas9.

Chemically synthesized gRNAs used in combination with ribonucleoprotein delivery hold promise for therapeutic application. While significant advancements have been made, challenges remain—particularly in understanding the mechanisms during gRNA-mediated cleavage and establishing robust methods for gRNA selection. Gaining insights into these processes is crucial for building an unbiased predictive model that can enhance genome editing efficiency and specificity. The manuscript submitted by Riesenberg et al. addresses these critical issues and has the potential for broad impact in the field. However, the contributions are often overstated and lack sufficient supporting results. Therefore, we cannot recommend publication of the manuscript in its current form and suggest the following revisions.

Major points

1) EVA activity score was trained on activity rates of 78 gRNAs, an insufficient number of datapoints for developing a robust and generalizable model. Additionally, the model's performance was evaluated using the same data on which it was trained (Fig. 3b), leading to an overestimation of its predictive capabilities and not accurately reflecting its real-world applicability. The manuscript defends the generalizability of the model by reporting Pearson correlations on 9 external data points (Line 164, Fig. 3c), a misleading metric given that Pearson correlations conventionally require at least 30 data points to be meaningful. When applied to published external datasets of sufficient size, the model performed poorly (Fig 4a), serving as evidence of model overfitting. To address these concerns, we recommend incorporating external datasets in model training to enhance statistical power and reduce the risk of overfitting.

2) Likewise, the model developed to predict rates of HDR efficiency requires more data for a meaningful evaluation; the manuscript misleadingly concludes that the model exhibits effective performance in external data by reporting Pearson correlations with 9 data points (Lines 186-188, Extended Data Fig. 5c).

3) According to Fig. 3f and accompanying text, '95% of human genome-wide gRNAs are predicted to be efficient when chemically synthesized' (Lines 167-168). This estimate suggests that random gRNA selection yields a suboptimal gRNA only 5% of the time, thus putting into question the utility of gRNA activity prediction. Additionally, this conclusion contradicts

the data presented in Fig. 4a, where EVA activity score exhibits a high rate of false positives. This inconsistency may arise from differences between in vivo gRNA activity—the metric on which the model was trained—and the metrics used in the published datasets. Further, the 95% estimate is largely an artifact of the liberal EVA threshold of 50 to define a guide as effective, serving as “an upper bound of the potential efficiency of a gRNA, i.e. if a gRNA has a bad EVA score (<50) it should also be inefficient irrespective of model system and gRNA production format used.” Benchmarking EVA score on synthetic gRNA performance alone instead of striving for a binary classifier that generalizes to transcribed gRNAs could enable EVA score to yield more informative predictions of synthetic gRNA activity.

4) The authors claimed that the “Conventional indel frequency metric can strongly underestimate gRNA activity” (Line 92), with the conventional metric being defined as indel quantification without the use of a DNA donor (Line 83). They defend this statement by reporting that, in contrast to reported gRNA activity from the conventional metric, accounting for cell death and enabling HDR shows that “a striking 93% of gRNAs have at least around 40% activity (mean 68%)” (Line 97). This evidence does not sufficiently defend their claim since it is not proven that the cell death is a direct artifact of gRNA activity, such as by demonstrating a lack of cell death with non-targeting guides. Without clear quantification of shortcomings or biases introduced with conventional methods, it is unclear how capturing alternative repair outcomes influences the construction of a superior dataset for model training. Additionally, it would be helpful to visualize the correlation between ‘Indels’ and ‘Activity’ directly, i.e. a scatter plot between the guides shown in red and blue in Figure 1c).

5) Authors frequently used Doench 2016 Rule Set 2 scores as the baseline comparison, but it would be more relevant to use the newer Cas9 activity prediction model, Rule Set 3 (DeWeirdt et al., 2022, PMID: 36068235).

6) The small scale of the data employed in this manuscript puts into question not only the validity of the models developed but also the general observations made. In Fig 2b, authors compared PAM-proximal dinucleotide gRNA efficiency metrics across in vivo transcribed gRNA and synthetic gRNA datasets. The probability of any specific dinucleotide at a given position is 6.25%. There are 78 guides generated in this study, meaning there are fewer than 5 guides on average for each dinucleotide. With such a small sample size, the statistical power of the analyses is insufficient to support any conclusions. Even the largest synthetic gRNA datasets in Fig. 2b yield on average 97 guides for each dinucleotide in contrast to over 4375 guides for each dinucleotide in transcribed gRNA activity data. Authors should address the limitations imposed by the dataset size, providing a more modest interpretation of the findings and acknowledging the constraints of the study.

7) The developed model was tested on several external datasets that report various gRNA efficiency metrics (as presented in Fig. 4), which is commendable. Including the information that was collated to perform this analysis within the extended data of this manuscript would provide a valuable resource for researchers interested in utilizing multiple of these datasets simultaneously.

Minor points

- 1) The Extended Data Fig. 4 legend reports two “c” sections.
- 2) The y-axis in Fig. 1c is difficult to interpret. Please provide explicit labels for each plot.
- 3) The consistent usage of color across the figures is generally helpful in building a clear narrative. However, we suggest changing the color of cell death in Fig. 1c and d, as pink is already used to indicate survival (color of the resazurin fluorescence assay) in Fig. 1b.
- 4) The interpretation of the two gray dotted lines on Fig. 5e is unclear.
- 5) Lines 70-71 (via our numbering... please use line numbers in future submissions to make peer review easier) overstate their contribution by claiming to “show that preventing perfect repair increases genome editing efficiency”, a hypothesis that is neither novel nor adequately supported in this manuscript.
- 6) Line 69 has a typo; “reliable” should be replaced with “reliably”.
- 7) The equations for death corrected NGS frequency (Lines 791-792), gRNA activity (Lines 794-795), and large scale repair correction factor for NGS frequency (Lines 808-810) have errors in formatting.
- 8) The title of Extended Data Figure 1c has a typo: “resazuring” should be replaced with “resazurin”.

Reviewer #2

(Remarks to the Author)

Ms. No.: NCOMMS-24-59699-T

Title: Robust prediction of synthetic gRNA activity and cryptic DNA repair by disentangling cellular CRISPR cleavage outcomes

First Author: Stephan Riesenber

Chemically synthesized gRNAs are preferred for many CRISPR-related medical applications due to their ability to incorporate chemical modifications and avoid sequence-dependent biases seen in transcribed gRNAs. They are not restricted by starting nucleotides and can be used in DNA-free gene editing, which is preferable for medical use. Existing tools for predicting gRNA activity often fail with synthetic gRNAs due to biases in transcription and differences in cleavage outcomes. Riesenber et al. described a new prediction model called EVA for prediction of chemically synthesized gRNA activity. EVA considers indels, targeted substitutions, and cell death and provides a more accurate measure of gRNA activity. Synthetic gRNAs show reduced impact from PAM-proximal bases, making them suitable for sequences with suboptimal nucleotides. The EVA activity score model reliably predicts synthetic gRNA activity and identifies inefficient gRNAs. For Cas12a, prediction tools trained on transcribed gRNAs also perform well with synthetic gRNAs. HDR efficiency

can be limited by DNA donor misfolding, and new models are being developed to predict HDR outcomes. Cas12a shows advantages over Cas9 in reducing unintended large-scale repairs, making it safer for gene therapy applications.

Although this manuscript would be of potentially broad interest, several points seem weak to me. Therefore, I think the manuscript is not strong enough in its current format and should be revised before publication.

Comments

1. They hybridize crRNA and tracrRNA and use them as gRNAs. However, all previous transcribed gRNA data used sgRNAs, which are covalently linked crRNAs and tracrRNAs. This difference may be the cause of the difference between chemically synthesized gRNAs and transcribed gRNAs that they claim. They should prove whether the same thing holds true for chemically synthesized sgRNAs.
2. "tracrRNA" should be corrected to "tracrRNA".
3. When creating a model, the model is usually created using the data set used for training, and then another data set of equal number is used to verify how valid the model is. The authors used 78 gRNAs for the training, however, they used only 9 gRNAs for verification. They should increase the number of data they use for validation.
4. How were the synthetic gRNAs used selected?
5. Wouldn't it be possible to verify more directly if the gRNA used in this screen was also tested with transcribed Cas9 gRNA?

Version 1:

Reviewer comments:

Reviewer #1

(Remarks to the Author)

While the authors addressed many of our concerns, some critical points remain unresolved that should be addressed prior to publication:

Comment #1

Our first major point expressed our concern of the generalizability of a model trained on 78 sgRNAs, suggesting the incorporation of external datasets in the training of the model.

The authors responded to this suggestion in the rebuttal by stating that "Because our score is based on a new metric that contains cell death and targeted substitutions in addition to conventionally used indels, it is not compatible for direct comparison or model training with published external datasets that quantify indels or phenotype depletion after a successful knockout."

We appreciate the fact that additional data may not be readily available, yet still this key deliverable of the manuscript may have poor generalizability due to its limited training set. This should be made clear to the reader.

Comment #2

Our third major point questions the real-world utility of this model given their estimate that 95% of synthetic guides are sufficiently active.

The authors responded to this comment by stating that "It is great that most gRNAs are efficient when provided in chemically synthesized form, but we believe it is valuable to be able to predict the rare, inefficient synthetic gRNAs, especially for projects where many targets need to be edited."

Given the presented estimation that inactive guides are so rare, a model to continuously predict gRNA activity has limited value. Thus, the merit of this model is its ability to identify the few that are inactive with an EVA score of <50, and the evidence of this presented in extended data Figure 7a should be moved to the main figures accordingly.

Comment #3:

Our third major point identifies that Figure 4a demonstrates excessive false positives associated with the model.

The authors responded by addressing the tendency of the EVA score to overestimate in vivo transcribed gRNA activity.

We would like the authors to address the high rate of false positives in the Leenay et al 2019 panel of Figure 4a, which reflects the performance of EVA activity score on synthetic gRNA data. While the authors speculate that false positives are an artifact of "cellular repair outcomes that are not accounted for" (Lines 206-207), namely the prevalence of gRNA activity that results in edits other than indels, Figure 1b of this manuscript reports that the majority of edits introduced by gRNAs result in indels. Thus, "outcomes that are not accounted for" alone are unlikely to explain the high abundance of false positives observed in this plot. Thus, the authors should address the possibility that these false positives also reflect the limitations of the model.

Reviewer #2

(Remarks to the Author)

The authors have fully addressed the points raised. Therefore, we are confident that this manuscript is of sufficient quality for publication in Nature communication.

Point-by-point response

Thank you for your valuable input on our manuscript 'Robust prediction of synthetic gRNA activity and cryptic DNA repair by disentangling cellular CRISPR cleavage outcomes'. We are very grateful for the time you have invested in improving our manuscript. Please find below a point-by-point response to your comments.

Reviewer #1

In this study, the authors presented three key results relevant to the prediction of synthetic Cas9 guide RNAs (gRNAs) activity. First, they screened 78 synthetic gRNAs with or without corresponding single-stranded DNA donors to detect evidence of gRNA activity from various DNA repair mechanisms. They further investigated the effect of including non-homologous end joining (NHEJ) inhibition using this set of gRNAs to infer the tendencies of perfect repair and large-scale repair. Second, they employed these data to assess features previously found to be associated with transcribed gRNA activity and develop the EVA activity score, a linear model with five features to predict synthetic gRNA activity. Following the model's development, the authors assessed the score on published datasets. Third, in response to EVA score overestimating gRNA activity when detected by homology directed repair (HDR), they developed a linear model for HDR efficiency prediction. The authors additionally reported that, despite suboptimal performance of models trained on transcribed Cas9 gRNA data to predict synthetic gRNA activity, models trained on transcribed Cas12a gRNA data yield more reliable predictions in the synthetic context, and identify direction for optimization. Moreover, they demonstrate that editing with Cas12a exhibits a stronger tendency towards perfect repair over large-scale repair as compared to Cas9.

Chemically synthesized gRNAs used in combination with ribonucleoprotein delivery hold promise for therapeutic application. While significant advancements have been made, challenges remain—particularly in understanding the mechanisms during gRNA-mediated cleavage and establishing robust methods for gRNA selection. Gaining insights into these processes is crucial for building an unbiased predictive model that can enhance genome editing efficiency and specificity. The manuscript submitted by Riesenberget al. addresses these critical issues and has the potential for broad impact in the field. However, the contributions are often overstated and lack sufficient supporting results. Therefore, we cannot recommend publication of the manuscript in its current form and suggest the following revisions.

Thank you for your kind words and constructive feedback. We have carefully considered your comments, addressed overstated claims, and provided additional supporting results to strengthen the manuscript.

Major points

- 1) EVA activity score was trained on activity rates of 78 gRNAs, an insufficient number of datapoints for developing a robust and generalizable model. Additionally, the model's performance was evaluated using the same data on which it was trained (Fig. 3b), leading to an overestimation of its predictive capabilities and not accurately reflecting its real-world applicability. The manuscript defends the generalizability of the model by reporting Pearson

correlations on 9 external data points (Line 164, Fig. 3c), a misleading metric given that Pearson correlations conventionally require at least 30 data points to be meaningful. When applied to published external datasets of sufficient size, the model performed poorly (Fig 4a), serving as evidence of model overfitting. To address these concerns, we recommend incorporating external datasets in model training to enhance statistical power and reduce the risk of overfitting.

Both you and reviewer 2 raised the important point of the necessity for increasing the number of gRNAs for the validation set. Based on your suggestion to have at least 30 data points for a meaningful correlation in the gRNA validation set, we have expanded the validation set to 40 targets (31 new gRNAs on top of the 9 original ones). With more gRNAs, the correlation is slightly lower than before ($r = 0.74$ vs. 0.81). We now write that the accuracy is 'robust' rather than 'comparable to the training set' as we had stated before:

'Then we validated the EVA score by determining the efficiency of forty gRNAs previously not used in model training. Importantly, we can predict the activity of those gRNAs with robust accuracy ($r = 0.74$) (Fig. 3d).'

Because our score is based on a new metric that contains cell death and targeted substitutions in addition to conventionally used indels, it is not compatible for direct comparison or model training with published external datasets that quantify indels or phenotype depletion after a successful knockout. Fig.4a shows that gRNA efficiencies from previous studies tend to be lower than predicted by the ascribed EVA scores. To interpret this, we write in the results that 'This could be due to cellular repair outcomes that are not accounted for, as well as sequence dependent transcription bias of transcribed gRNAs, which both tend to underestimate gRNA activity in these published datasets'. We now also added a sentence in the discussion: 'The EVA score is often much higher than the measured efficiencies in published datasets, which does not seem to be due to poor performance of the score, but rather to better performance of gRNAs when delivered in chemically synthesized form, as the four worst published transcribed gRNAs from the largest dataset all have more than 79% synthetic gRNA activity (Fig. 4c and d, Extended Data Fig. 6).'

In Fig.4c,d and new Extended Data Fig. 6 we attempt to show that the sometime large difference of EVA scores to measured efficiencies for transcribed gRNAs from other datasets is not an artefact of overfitting, but is due to differences in transcribed and synthetic gRNAs. The four worst measured gRNAs from the Doench 2016 dataset have a high EVA score and also have a high experimentally determined *in vivo* activity when supplied as synthetic gRNAs.

We conclude that when comparing to datasets that report a different efficiency metric 'a more meaningful way to benchmark our score would be assessing its ability to serve as an upper bound of the potential efficiency of a gRNA'.

2) Likewise, the model developed to predict rates of HDR efficiency requires more data for a meaningful evaluation; the manuscript misleadingly concludes that the model exhibits effective performance in external data by reporting Pearson correlations with 9 data points (Lines 186-188, Extended Data Fig. 5c).

As for the first comment we used the same 31 new additional targets for an updated HDR prediction score validation and achieved similar correlations to the measured HDR efficiency as before ($r = 0.59$ vs $r = 0.58$) (Extended Data Fig. 5d).

We were surprised to see that among the new targets was one (*TTF1*) that resulted in very high HDR efficiency, which can normally only be achieved by inhibiting competing DNA repair pathways. In the discussion we now write: ‘Excitingly, when DNA donor sequence characteristics are optimal, an exceptional HDR efficiency of up to 78% can be achieved (Extended Data Fig. 5d).’

3) According to Fig. 3f and accompanying text, ‘95% of human genome-wide gRNAs are predicted to be efficient when chemically synthesized’ (Lines 167-168). This estimate suggests that random gRNA selection yields a suboptimal gRNA only 5% of the time, thus putting into question the utility of gRNA activity prediction. Additionally, this conclusion contradicts the data presented in Fig. 4a, where EVA activity score exhibits a high rate of false positives. This inconsistency may arise from differences between *in vivo* gRNA activity—the metric on which the model was trained—and the metrics used in the published datasets. Further, the 95% estimate is largely an artifact of the liberal EVA threshold of 50 to define a guide as effective, serving as “an upper bound of the potential efficiency of a gRNA, i.e. if a gRNA has a bad EVA score (<50) it should also be inefficient irrespective of model system and gRNA production format used.” Benchmarking EVA score on synthetic gRNA performance alone instead of striving for a binary classifier that generalizes to transcribed gRNAs could enable EVA score to yield more informative predictions of synthetic gRNA activity.

It is great that most gRNAs are efficient when provided in chemically synthesized from, but we believe it is valuable to be able to predict the rare, inefficient synthetic gRNAs, especially for projects where many targets need to be edited. As discussed in our answer to point one, we agree that inconsistencies arise from differences between *in vivo* gRNA activity—the metric on which the model was trained—and the metrics used in the published datasets. In Fig.4c,d and new Extended Data Fig. 6 we attempt to show that the EVA score does not have a high rate of ‘false positives’. The four worst measured gRNAs from the Doench 2016 dataset have a high EVA score and also have a high experimentally determined *in vivo* activity when supplied as synthetic gRNAs.

While we think there is value in a binary classifier for ease of application (EVA score <50 bad, EVA score ≥50 good), we agree that it adds value to also present more fine-grained benchmarking. The original USCS genome browser track for CRISPR gRNAs has a ternary classifier (bad, moderate, good), but this was made for transcribed gRNAs, whose efficiencies are much more variable. The USCS genome browser tracks for the EVA scores provided with this study are not only color-coded by the binary classifier, but also show the numerical score on the left. For easy comparison to transcribed gRNA prediction scores we added tracks for the Rule Set 3 Hsu and Chen tracrRNA after your suggestion to compare to Rule Set 3 as baseline (see point 5).

We now also updated the pie charts of Fig.3f,g and Extended Data Fig. 7 with more EVA score bins (now differentiating between <50, ≥50-60, >60-70, >70-80,>80).

4) The authors claimed that the “Conventional indel frequency metric can strongly underestimate gRNA activity” (Line 92), with the conventional metric being defined as indel quantification without the use of a DNA donor (Line 83). They defend this statement by reporting that, in contrast to reported gRNA activity from the conventional metric, accounting for cell death and enabling HDR shows that “a striking 93% of gRNAs have at least around 40% activity (mean 68%)” (Line 97). This evidence does not sufficiently defend their claim since it is not proven that the cell death is a direct artifact of gRNA activity, such as by

demonstrating a lack of cell death with non-targeting guides. Without clear quantification of shortcomings or biases introduced with conventional methods, it is unclear how capturing alternative repair outcomes influences the construction of a superior dataset for model training. Additionally, it would be helpful to visualize the correlation between 'Indels' and 'Activity' directly, i.e. a scatter plot between the guides shown in red and blue in Figure 1c).

We did the suggested experiments to show this and now write: 'Non-targeting gRNAs do not reduce cell survival (Extended Data Fig. 2f), while cell death introduced by targeting gRNAs can result in big differences of indels and *in vivo* gRNA activity (Extended Data Fig. 2g).'

5) Authors frequently used Doench 2016 Rule Set 2 scores as the baseline comparison, but it would be more relevant to use the newer Cas9 activity prediction model, Rule Set 3 (DeWeirdt et al., 2022, PMID: 36068235).

Thank you for this suggestion. We now incorporate comparisons to Rule Set 3 in Fig. 2, 3, Extended Data Fig. 6, Extended Data Fig. 7. Indeed, Rule Set 3 with the Chen tracrRNA performs better in predicting *in vivo* gRNA activity (Fig. 2a: $r = 0.42$) than the other published scores, and combining both the Hsu and Chen tracrRNA design predictions also results in a higher percentage of correctly identified worst 25% (Q1) gRNAs for several datasets – compared to Rule Set 2 predictions as used before.

As the original DeWeirdt et al. paper only provided a Python script for Rule Set 3 calculation - which is not easily accessible for researchers without bioinformatics skills - we precalculated the scores for both the Hsu and the Chen tracrRNA for the whole human genome and provide them as USCS genome browser tracks for easy web browser access.

6) The small scale of the data employed in this manuscript puts into question not only the validity of the models developed but also the general observations made. In Fig 2b, authors compared PAM-proximal dinucleotide gRNA efficiency metrics across *in vivo* transcribed gRNA and synthetic gRNA datasets. The probability of any specific dinucleotide at a given position is 6.25%. There are 78 guides generated in this study, meaning there are fewer than 5 guides on average for each dinucleotide. With such a small sample size, the statistical power of the analyses is insufficient to support any conclusions. Even the largest synthetic gRNA datasets in Fig. 2b yield on average 97 guides for each dinucleotide in contrast to over 4375 guides for each dinucleotide in transcribed gRNA activity data. Authors should address the limitations imposed by the dataset size, providing a more modest interpretation of the findings and acknowledging the constraints of the study.

To acknowledge the constraints, we now write in the results: 'Of note, the dataset for transcribed sgRNAs is more than 40-fold larger than the biggest synthetic gRNA dataset. Due to comparatively small sample sizes for each dinucleotide in studies using synthetic gRNAs, we cannot exclude potential moderate effects of PAM-proximal dinucleotides on synthetic gRNA efficiency.' and in the discussion: 'Contrary to transcribed gRNAs, synthetic gRNAs show a strongly reduced impact of PAM-proximal bases on activity (Fig. 2b), albeit more data is needed to accurately quantify their impact in synthetic gRNAs.'

7) The developed model was tested on several external datasets that report various gRNA efficiency metrics (as presented in Fig. 4), which is commendable. Including the information that was collated to perform this analysis within the extended data of this manuscript would

provide a valuable resource for researchers interested in utilizing multiple of these datasets simultaneously.

We now made a new Supplementary Data file (Supplementary Data 4) that contains the different gRNA efficiency datasets. We now also provide Source Data files for each figure.

Minor points

1) The Extended Data Fig. 4 legend reports two “c” sections.

We have corrected this.

2) The y-axis in Fig. 1c is difficult to interpret. Please provide explicit labels for each plot.

We provided explicit labels for each y-axis.

3) The consistent usage of color across the figures is generally helpful in building a clear narrative. However, we suggest changing the color of cell death in Fig. 1c and d, as pink is already used to indicate survival (color of the resazurin fluorescence assay) in Fig. 1b.

Thank you for pointing this out. We now use purple as the color of cell death throughout the paper.

4) The interpretation of the two gray dotted lines on Fig. 5e is unclear.

To clarify this, we have added the following sentence in the figure legend for 5e: ‘The dotted lines between the resazurin assay panel and the Illumina sequencing panel indicate that only DNA extracted from surviving cells will be sequenced.’

5) Lines 70-71 (via our numbering... please use line numbers in future submissions to make peer review easier) overstate their contribution by claiming to “show that preventing perfect repair increases genome editing efficiency”, a hypothesis that is neither novel nor adequately supported in this manuscript.

To avoid overstatement of our contribution we have removed this claim from the end of the introduction and now write ‘In addition, we show ~~that preventing perfect repair increases genome editing efficiency and~~ that Cas9 and Cas12a have different propensities for cellular cleavage outcomes.’ We have also added line numbers to the manuscript to make peer review easier.

6) Line 69 has a typo; “reliable” should be replaced with “reliably”.

We have corrected the typo.

7) The equations for death corrected NGS frequency (Lines 791-792), gRNA activity (Lines 794-795), and large scale repair correction factor for NGS frequency (Lines 808-810) have errors in formatting.

We have updated the formatting.

8) The title of Extended Data Figure 1c has a typo: “resazuring” should be replaced with “resazurin”.

We have corrected the typo.

Reviewer #2

Chemically synthesized gRNAs are preferred for many CRISPR-related medical applications due to their ability to incorporate chemical modifications and avoid sequence-dependent biases seen in transcribed gRNAs. They are not restricted by starting nucleotides and can be used in DNA-free gene editing, which is preferable for medical use. Existing tools for predicting gRNA activity often fail with synthetic gRNAs due to biases in transcription and differences in cleavage outcomes. Riesenber et al. described a new prediction model called EVA for prediction of chemically synthesized gRNA activity. EVA considers indels, targeted substitutions, and cell death and provides a more accurate measure of gRNA activity. Synthetic gRNAs show reduced impact from PAM-proximal bases, making them suitable for sequences with suboptimal nucleotides. The EVA activity score model reliably predicts synthetic gRNA activity and identifies inefficient gRNAs. For Cas12a, prediction tools trained on transcribed gRNAs also perform well with synthetic gRNAs. HDR efficiency can be limited by DNA donor misfolding, and new models are being developed to predict HDR outcomes. Cas12a shows advantages over Cas9 in reducing unintended large-scale repairs, making it safer for gene therapy applications.

Although this manuscript would be of potentially broad interest, several points seem weak to me. Therefore, I think the manuscript is not strong enough in its current format and should be revised before publication.

Thank you for your valuable feedback. We have addressed each point below and revised the manuscript accordingly to strengthen its quality and clarity.

Comments

1. They hybridize crRNA and tracrRNA and use them as gRNAs. However, all previous transcribed gRNA data used sgRNAs, which are covalently linked crRNAs and tracrRNAs. This difference may be the cause of the difference between chemically synthesized gRNAs and transcribed gRNAs that they claim. They should prove whether the same thing holds true for chemically synthesized sgRNAs.

This is an excellent point. To address this, we experimentally compared the gRNA activity for eight targets with synthetic hybridized crRNA:tracrRNA, synthetic sgRNA, and also sgRNA encoded in dsDNA and transcribed with a U6 promoter (not chemically synthesized). We have now added the related data in the new Extended Data Fig. 6 'Comparison of gRNA architectures and delivery forms'. In the results section we write 'When directly comparing different gRNA architectures and delivery forms in 409B2 hiPSCs we observe comparable efficient *in vivo* activities for all eight targets for both synthetic hybridized crRNA:tracrRNA and synthetic sgRNA, while sgRNAs encoded in exogenous dsDNA and transcribed from a U6

promoter are less efficient for the previously described inefficient transcribed gRNAs from the Doench *et al.* 2016 dataset (Extended Data Fig. 6).'

2. "tracrRNA" should be corrected to "tracrRNA".

We have corrected the typo.

3. When creating a model, the model is usually created using the data set used for training, and then another data set of equal number is used to verify how valid the model is. The authors used 78 gRNAs for the training, however, they used only 9 gRNAs for verification. They should increase the number of data they use for validation.

Reviewer 1 also stated that we should test more gRNAs for validation and pointed out that Pearson correlations require at least 30 data points to be meaningful. To provide a meaningful gRNA validation set, we have expanded the set to 40 targets (31 new gRNAs on top of the 9 original ones). With more gRNAs, the correlation is slightly lower than before ($r = 0.74$ vs. 0.81). We now write that the accuracy is 'robust' rather than 'comparable to the training set' as we had stated before:

'Then we validated the EVA score by determining the efficiency of forty gRNAs previously not used in model training. Importantly, we can predict the activity of those gRNAs with robust accuracy ($r = 0.74$) (Fig. 3d).'

We also used the same additional targets in an updated HDR prediction score validation and achieved similar correlations to the measured HDR efficiency as before ($r = 0.59$ vs $r = 0.58$) (Extended Data Fig. 5d).

4. How were the synthetic gRNAs used selected?

We have added following sentence in the methods section: 'For the screen, we used gRNAs from previous and planned evolutionary studies, mainly selected to cut close to genomic sites where modern humans and archaic humans differ.'

5. Wouldn't it be possible to verify more directly if the gRNA used in this screen was also tested with transcribed Cas9 gRNA?

In our answer to your first point we also included a direct comparison of eight synthetic gRNAs to their respective transcribed gRNAs. Our own experimental data of transcribed gRNAs is comparable to Rule Set 3 predictions (updated Doench score - DeWeirdt *et al.*, 2022, PMID: 36068235), and recapitulates the published tendency of efficiency differences. In the results section we write 'When directly comparing different gRNA architectures and delivery forms in 409B2 hiPSCs we observe comparable efficient *in vivo* activities for all eight targets for both synthetic hybridized crRNA:tracrRNA and synthetic sgRNA, while sgRNAs encoded in exogenous dsDNA and transcribed from a U6 promoter are less efficient for the previously described inefficient transcribed gRNAs from the Doench *et al.* 2016 dataset (Extended Data Fig. 6).'

Point-by-point response

Please find below a point-by-point response to your comments. Thank you again for your input.

Reviewer #1

While the authors addressed many of our concerns, some critical points remain unresolved that should be addressed prior to publication:

Comment #1

Our first major point expressed our concern of the generalizability of a model trained on 78 sgRNAs, suggesting the incorporation of external datasets in the training of the model.

The authors responded to this suggestion in the rebuttal by stating that “Because our score is based on a new metric that contains cell death and targeted substitutions in addition to conventionally used indels, it is not compatible for direct comparison or model training with published external datasets that quantify indels or phenotype depletion after a successful knockout.”

We appreciate the fact that additional data may not be readily available, yet still this key deliverable of the manuscript may have poor generalizability due to its limited training set. This should be made clear to the reader.

We now write in the discussion: ‘However, more *in vivo* gRNA activity data from additional gRNAs could improve future models, as the current number used to train our model may limit generalizability.’

Comment #2

Our third major point questions the real-world utility of this model given their estimate that 95% of synthetic guides are sufficiently active.

The authors responded to this comment by stating that “It is great that most gRNAs are efficient when provided in chemically synthesized form, but we believe it is valuable to be able to predict the rare, inefficient synthetic gRNAs, especially for projects where many targets need to be edited.”

Given the presented estimation that inactive guides are so rare, a model to continuously predict gRNA activity has limited value. Thus, the merit of this model is its ability to identify the few that are inactive with an EVA score of <50, and the evidence of this presented in extended data Figure 7a should be moved to the main figures accordingly.

We have moved the data from Extended Data Figure 7a to Figure 4 (new panel f).

Comment #3:

Our third major point identifies that Figure 4a demonstrates excessive false positives associated with the model.

The authors responded by addressing the tendency of the EVA score to overestimate in vivo transcribed gRNA activity.

We would like the authors to address the high rate of false positives in the Leenay et al 2019 panel of Figure 4a, which reflects the performance of EVA activity score on synthetic gRNA data. While the authors speculate that false positives are an artifact of “cellular repair outcomes that are not accounted for” (Lines 206-207), namely the prevalence of gRNA activity that results in edits other than indels, Figure 1b of this manuscript reports that the majority of edits introduced by gRNAs result in indels. Thus, “outcomes that are not accounted for” alone are unlikely to explain the high abundance of false positives observed in this plot. Thus, the authors should address the possibility that these false positives also reflect the limitations of the model.

We now write in the discussion: ‘Even though the synthetic gRNA dataset from Leenay *et al.* is free of gRNA transcription bias, EVA scores are still often much higher than the measured indels (Fig. 4a). While this could be due to false positives, which would reflect a limitation of the EVA score, few measured indels and a high EVA score would also be compatible with high gRNA cleavage induced death of the primary human T cells used in their study, which are uniquely susceptible to DNA damage (PMID: 28533414).’

Reviewer #2

The authors have fully addressed the points raised. Therefore, we are confident that this manuscript is of sufficient quality for publication in Nature communication.

We are happy to hear this.